# In vitro γ-aminobutyric acid A (GABA$_A$) receptor activity and binding interactions at the α$^+$/γ$_2$$^-$ interface of 53 prescription and designer benzodiazepines

Caitlyn Norman [1] ✉, Sara I. Liin [2], Amaia Jauregi-Miguel[2,3], Nina E. Ottosson[2,3] & Henrik Gréen [1,4]

Although widely prescribed globally, benzodiazepines (BZDs) are commonly misused and involved in a high rate of drug-related deaths worldwide. Novel chemically similar compounds, designer BZDs (DBZDs), have emerged on the recreational market with little information available on their pharmacodynamic effects. An in vitro α$_1$β$_2$γ$_2$ GABA$_A$ assay was used to determine the activity and explore structure-activity relationships of 42 DBZDs plus 11 prescription BZDs. Their interaction with the BZD α$^+$/γ$_2$$^-$ interface was explored using the BZD antagonist flumazenil. Most of the D/BZDs were positive allosteric modulators, but 4'-chlorodiazepam and 4'-fluorodiazepam were negative allosteric modulators, although the activity was not antagonized by flumazenil. All metabolites tested were active. This data informs the interpretation of intoxications, harm reduction measures, and drugs legislation for DBZDs; demonstrates the importance of testing new drugs as they emerge on the recreational market; and offers insights into how BZD chemical modifications alter GABA$_A$ pharmacodynamics.

Benzodiazepines (BZDs) are anxiolytic and hypnotic drugs that are widely prescribed globally for conditions including anxiety, insomnia, and epilepsy[1]. BZDs derive their names from their chemical structure, which consists of a benzene ring fused to a heterocyclic diazepine ring, as can be seen in the traditional 1,4-benzodiazepine and 1,3,4-triazolobenzodiazepine base structures in Tables 1A and 1B, respectively. However, since 2007, novel chemically similar compounds designed to mimic the clinical and pharmacological properties of BZDs have emerged on the recreational drugs market, which are often referred to as "new psychoactive substance (NPS) BZDs", "novel BZDs", and "designer BZDs (DBZDs)"[1]. Going forward, the term "DBZDs" will be used. Most DBZDs have 1,4-BZD or 1,3,4-triazoloBZD base structures, but some have different base structures that fall outside of the chemical definition of a BZD, such as the thienotriazolodiazepines (e.g., etizolam (44)), which consist of a thieno ring fused to a heterocyclic diazepine ring (Table 1D)[2].

BZDs are positive allosteric modulators (PAMs) of the γ-aminobutyric acid type A (GABA$_A$) neurotransmitter receptors[3]. GABA$_A$ receptors are ligand-gated chloride channels that consist of a combination of five different subunits (pentameric), where BZDs have been found to bind to the α$^+$/γ2$^-$ interfaces and thereby enhance GABA-induced chloride ion flux, leading to neuronal hyperpolarization, as depicted in Fig. 1A[4,5]. Despite their differing structures, DBZDs are assumed to also bind to the α$^+$/γ2$^-$ interfaces, but it has not yet been determined experimentally. However, recent studies have found that some, but not all BZDs bind to two transmembrane β$^+$/α$^-$ interfaces, known to selectively bind the anesthetic etomidate, but with lower efficacy[6–9]. One study using in silico modelling also found diazepam (13) may bind to the γ$^+$/β$^-$ interface, one of the barbiturate binding sites[10]. This indicates that there may also be differences in the binding interfaces of DBZDs, which could change the pharmacodynamic effects of the drugs.

The effects of DBZDs commonly reported by people using them recreationally include hypnosis, heavy sedation, long-lasting amnesia, anxiolysis, euphoria, loss of control, severe withdrawal, and rapid development of tolerance[11]. There are a variety of reasons for the recreational use of DBZDs, such as the self-medication of anxiety or sleeping disorders[2]; however, they are most often abused in conjunction with other drugs, particularly opiates and alcohol, in order to alleviate withdrawal symptoms,

[1]Division of Clinical Chemistry and Pharmacology, Department of Biomedical and Clinical Sciences, Faculty of Medicine, Linköping University, Linköping, Sweden. [2]Division of Neurobiology, Department of Biomedical and Clinical Sciences, Linköping University, Linköping, Sweden. [3]Chemical Biology Consortium Sweden, Science for Life Laboratory, Linköping University, Linköping, Sweden. [4]Department of Forensic Genetics and Forensic Toxicology, National Board of Forensic Medicine, Linköping, Sweden. ✉e-mail: caitlyn.norman@liu.se

**Table 1 | Benzodiazepines (BZDs) included in this study with their structural components and year notified by the European Union Early Warning System (EU EWS)**

| Base | Name | R₁ | R₂ | R₃ | R₄ | R₅ | R₆ | R₇ | R₈ | Year |
|---|---|---|---|---|---|---|---|---|---|---|
| A | **(1)** 2'-bromodiazepam | CH₃ | - | H | Cl | - | Br | H | H | - |
| | **(2)** 2'-bromonordazepam (2'-bromonordiazepam) | H | - | H | Cl | - | Br | H | H | - |
| | **(3)** 2,4'-dichlorodiazepam (Ro5-6900) | CH₃ | - | H | Cl | - | Cl | Cl | H | - |
| | **(4)** 3-hydroxyphenazepam | H | - | OH | Br | - | Cl | H | H | 2016 |
| | **(5)** 4'-chlorodiazepam (Ro5-4864) | CH₃ | - | H | Cl | - | H | Cl | H | 2016 |
| | **(6)** 4'-fluorodiazepam | CH3 | - | H | Cl | - | H | F | H | - |
| | **(7)** Cinazepam | H | - | OOC(CH₂)₂COOH | Br | - | Cl | H | H | 2019 |
| | **(8)** Clonazepam (Klonopin; Ro5-4023) | H | - | H | NO₂ | - | Cl | H | H | - |
| | **(9)** Cloniprazepam (Ro5-4200) | CH₃C₃H₄ | - | H | NO₂ | - | Cl | H | H | 2016 |
| | **(10)** Delorazepam (nordiclazepam) | H | - | H | Cl | - | Cl | H | H | - |
| | **(11)** Desalkylgidazepam | H | - | H | Br | - | H | H | H | 2022 |
| | **(12)** Desmethylflunitrazepam (fonazepam; Ro5-4435) | H | - | H | NO₂ | - | F | H | H | 2016 |
| | **(13)** Diazepam (Valium; Ro5-2807) | CH₃ | - | H | Cl | - | H | H | H | - |
| | **(14)** Diclazepam (2'-chlorodiazepam; Ro5-3448) | CH₃ | - | H | Cl | - | Cl | H | H | 2013 |
| | **(15)** Difludiazepam (Ro07-4065) | CH₃ | - | H | Cl | - | F | H | F | 2017 |
| | **(16)** Flubromazepam | H | - | H | Br | - | F | H | H | 2013 |
| | **(17)** Fludiazepam (Ro5-3438) | CH₃ | - | H | Cl | - | F | H | H | - |
| | **(18)** Gidazepam | CH₂CONHNH₂ | - | H | Br | - | H | H | H | 2024 |

**Table 1 (continued) | Benzodiazepines (BZDs) included in this study with their structural components and year notified by the European Union Early Warning System (EU EWS)**

| Base | Name | Structural components | | | | | | | | | Year |
|------|------|-------|-------|-------|-------|-------|-------|-------|-------|-------|
| | | $R_1$ | $R_2$ | $R_3$ | $R_4$ | $R_5$ | $R_6$ | $R_7$ | $R_8$ | |
| | (19) Lorazepam (Ativan) | H | - | OH | Cl | - | Cl | H | H | - |
| | (20) Meclonazepam (Ro11-3128) | H | - | CH$_3$ | NO$_2$ | - | Cl | H | H | 2014 |
| | (21) Methylclonazepam (Ro05-4082) | CH$_3$ | - | H | NO$_2$ | - | Cl | H | H | 2018 |
| | (22) Nifoxipam (3-hydroxy-desmethylflunitrazepam) | H | - | OH | NO$_2$ | - | F | H | H | 2015 |
| | (23) Nitrazepam | H | - | H | NO$_2$ | - | H | H | H | - |
| | (24) Nordazepam (nordiazepam; Ro5-2180) | H | - | H | Cl | - | H | H | H | - |
| | (25) Norfludiazepam (desalkylflurazepam; Ro5-3367) | H | - | H | Cl | - | F | H | H | 2017 |
| | (26) Phenazepam | H | - | H | Br | - | Cl | H | H | 2007 |
| | (27) Temazepam (Ro5-5345; Ro5-5354) | CH$_3$ | - | OH | Cl | - | H | H | H | - |
| B | (28) 4'-chloro deschloroalprazolam | - | CH$_3$ | H | H | - | H | H | H | 2023 |
| | (29) Adinazolam | - | CH$_2$N(CH$_3$)$_2$ | H | Cl | - | H | H | H | 2015 |
| | (30) Alprazolam (Xanax) | - | CH$_3$ | H | Cl | - | H | H | H | - |
| | (31) Bromazolam | - | CH$_3$ | H | Br | - | H | H | H | 2016 |
| | (32) Clobromazolam (phenazolam) | - | CH$_3$ | H | Br | - | Cl | H | H | 2018 |
| | (33) Clonazolam (clonitrazolam) | - | CH$_3$ | H | NO$_2$ | - | Cl | H | H | 2015 |
| | (34) Flualprazolam | - | CH$_3$ | H | Cl | - | F | H | H | 2018 |

**Table 1 (continued) | Benzodiazepines (BZDs) included in this study with their structural components and year notified by the European Union Early Warning System (EU EWS)**

| Base | Name | Structural components | | | | | | | | | Year |
|---|---|---|---|---|---|---|---|---|---|---|---|
| | | $R_1$ | $R_2$ | $R_3$ | $R_4$ | $R_5$ | $R_6$ | $R_7$ | $R_8$ | | |
| | (35) Flubromazolam | - | $CH_3$ | H | Br | - | F | H | H | | 2014 |
| | (36) Flunitrazolam | - | $CH_3$ | H | $NO_2$ | - | F | H | H | | 2016 |
| | (37) Nitrazolam | - | $CH_3$ | H | $NO_2$ | - | H | H | H | | 2015 |
| | (38) Pyrazolam | - | $CH_3$ | H | Br | - | NH on ring | H | H | | 2012 |
| C | (39) Rilmazolam | - | $CON(CH_3)_2$ | H | Cl | - | Cl | H | H | | 2025 |
| D | (40) Brotizolam | - | $CH_3$ | - | - | Br | Cl | H | H | | - |
| | (41) Clotizolam (Ro11-1465) | - | $CH_3$ | - | - | Cl | Cl | H | H | | - |
| | (42) Deschloroclotizolam | - | $CH_3$ | - | - | Cl | H | H | H | | 2021 |
| | (43) Deschloroetizolam | - | $CH_3$ | - | - | $CH_2CH_3$ | H | H | H | | 2014 |
| | (44) Etizolam | - | $CH_3$ | - | - | $CH_2CH_3$ | Cl | H | H | | 2011 |
| | (45) Flubrotizolam | - | $CH_3$ | - | - | Br | F | H | H | | 2021 |
| | (46) Fluclotizolam | - | $CH_3$ | - | - | Cl | F | H | H | | 2018 |
| | (47) Fluetizolam | - | $CH_3$ | - | - | $CH_2CH_3$ | F | H | H | | 2022 |
| | (48) Metizolam (desmethyletizolam) | - | H | - | - | $CH_2CH_3$ | Cl | H | H | | 2015 |

Structures of the four main base groups are provided at the top: (A) 1,4-BZDs, (B) 1,3,4-triazoloBZDs, (C) 1,2,4-triazoloBZDs, and (D) thienotriazoloBZDs. Five BZDs included in this study had other base groups, so their structures are provided separately: (E) thionordiazepam (49), a 1,4-BZD with a sulfur group, notified to the EU EWS in 2017; (F) tofisopam (50), a 2,3-BZD notified to the EU EWS in 2018; (G) bretazenil (51), an imidazopyrroloBZD notified to the EU EWS in 2021; (H) bentazepam (52), a thienoBZD notified to the EU EWS in 2019; (I) rilmazafone (53), an open-ring 1,2,4-triazoloBZD notified to the EU EWS in 2022; and (J) flumazenil (54), a pyrroloBZD and BZD antagonist.

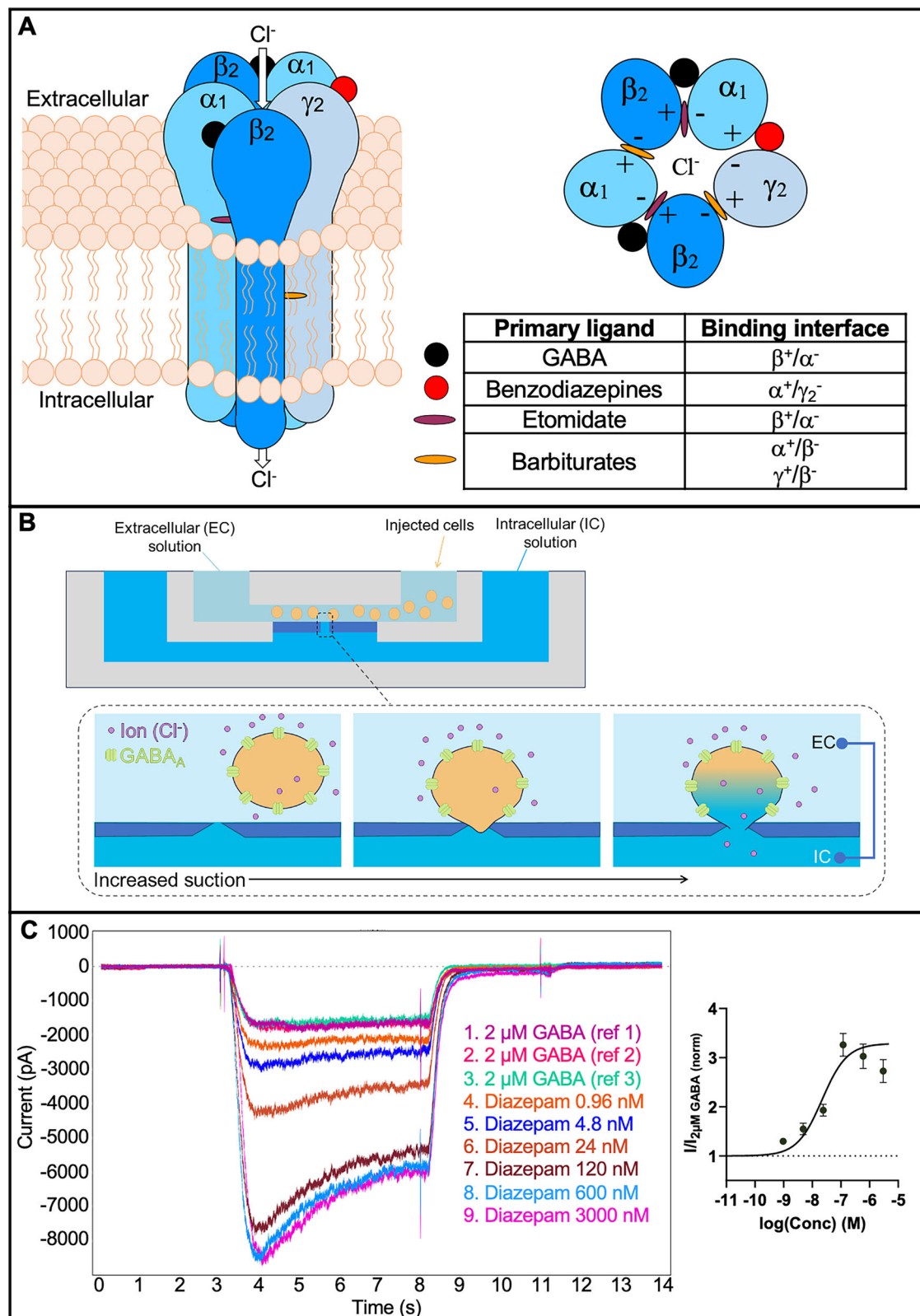

**Fig. 1 | Depiction of the $\alpha_1\beta_2\gamma_2$ GABA$_A$ receptor assay used in this study.**
**A** Representation of the $\alpha_1\beta_2\gamma_2$ GABA$_A$ receptor structure (a chloride ion-permeable channel formed by a combination of five different subunits) with the primary binding interfaces for different ligands, including GABA, benzodiazepines, etomidate, and barbiturates. **B** Depiction of a well with integrated flow channels and electrodes for the automated planar patch-clamp on mammalian Chinese hamster ovary (CHO) cells stably expressing the $\alpha_1\beta_2\gamma_2$ GABA$_A$ receptor used in this study.

Intracellular solution is injected into one flow channel, and extracellular solution and then cells are injected into another flow channel. Increased suction is applied to pull the cells into a hole and rupture the cell membrane to provide access to the interior and exterior of the cell. **C** The electrical current from one cell for diazepam **(13)** generated in this study and the resulting concentration-response curve for diazepam **(13)** from a minimum of four independent experiments ($n \geq 4$).

enhance the euphoric effects of the other drug(s), and reduce the unwanted effects of the other drug(s), such as insomnia from the use of stimulants[12,13].

BZDs are commonly detected in drug-related deaths in countries around the world, with DBZDs from illicitly produced tablets and blotters increasingly implicated[14–18]. For example, in 2022, BZDs were involved in 40-75% of drug-related deaths in Australia, Austria, Denmark, Finland, Luxembourg, Portugal, Scotland, and Slovenia[14,17,19]. Previously, the majority of the illicit BZD tablets circulating in the European market were diverted prescription BZDs, particularly diazepam (**13**; Valium) and alprazolam (**30**; Xanax). However, more recently, there has been an increase in the availability of illicitly produced tablets and blotters that are often indistinguishable from their pharmaceutical counterparts, leading to unpredictability for consumers with regard to the specific drug, potency, or dose of the drug they are consuming[1,20]. Due to the unpredictability of the drugs on the illicit market, increasing use of DBZDs since their emergence in 2007, and severe risks of harm, the use of DBZDs is recognized as a global threat to public health by the United Nations Office on Drugs and Crime (UNODC)[21].

DBZDs are typically close analogs of prescription BZDs or other DBZDs with certain common areas of substitution, as can be seen in Table 1, where the addition or removal of methyl groups, halogens (fluorine, chlorine, or bromine), or nitro groups are the most common substitutions. Similar to other classes of NPS, many DBZDs have been pulled from published research studies or patents documenting their synthesis, such as from Hoffman-La Roche in the 1960s and 1970s[22]. Literature available on the metabolism of BZDs may also be used by clandestine producers as a source of potential DBZDs to synthesize, since active metabolites of prescription BZDs or DBZDs have emerged on the illicit market (e.g., 3-hydroxyphenazepam (**4**), desmethylflunitrazepam (**12**; fonazepam))[23]. Prodrugs of previously available BZDs (e.g., cloniprazepam (**9**)) have also emerged and been implicated in criminal cases[24]. One of the most common types of prodrugs to emerge is benzophenones, which are ring-open derivatives of BZDs, such as rilmazafone (**53**) (Table 1I), that rapidly metabolize into the corresponding BZDs (e.g., rilmazolam (**39**) for rilmazafone (**53**))[25,26].

There is a lack of pharmacological data available on these drugs, although it has been reported anecdotally that DBZDs vary greatly in their potency, onset times, and pharmacological half-lives[1,20]. Some of the DBZDs are approved for medical use in a select number of countries, often Russia, India, Italy, and/or Japan. For example, the first DBZD to be reported on the recreational market in the EU, phenazepam (**26**)[27], and one of the most recent to emerge, gidazepam (**18**), are prescription drugs only available in Russia, as well as Ukraine for gidazepam (**18**)[28,29]. This means there is more pharmacological data available on these drugs, but the data are often only published in the native language (e.g., Russian, Japanese) and are thereby not easily accessible to researchers in other countries. Other compounds that have emerged on the illicit market, such as flubromazolam (**35**) and bromazolam (**31**), have never undergone clinical trials, so data on their relative effects and harms remain limited within the scientific literature. The lack of pharmacological data on DBZDs makes it a challenge to understand their potential harm to consumers and provide appropriate harm reduction advice, such as dosing guidelines, to people who use drugs.

In this study, an in vitro $\alpha_1\beta_2\gamma_2$ GABA$_A$ receptor activity assay was developed and used to examine the activity of 42 DBZDs plus 11 prescription BZDs, including all but one of the DBZDs reported to the European Union Drugs Agency (EUDA) through April 2025. In addition, the extent of the activity of the DBZDs coming from binding at the $\alpha^+/\gamma2^-$ interface was evaluated by applying the BZD antagonist, flumazenil (**54**), to the assay to block the $\alpha^+/\gamma2^-$ interface.

## Results

### Large variation observed in the in vitro GABA$_A$ receptor activity of 53 BZDs

To assess the in vitro GABA$_A$ receptor activity of 53 compounds, including 42 DBZDs and 11 prescription BZDs, an automated planar patch-clamp technique on mammalian Chinese hamster ovary (CHO) cells stably expressing the $\alpha_1\beta_2\gamma_2$ GABA$_A$ receptor was used (see "Methods" for more

information). An example of the electrical currents recorded with this method from one cell for increasing concentrations of diazepam is shown in Fig. 1B, along with the resulting concentration-response curve for diazepam from a minimum of four independent experiments ($n \geq 4$). All but one of the DBZDs reported to the EUDA through April 2025 were examined. The "alprazolam precursor" ((2-(3-(aminomethyl)-5-methyl-4$H$-1,2,4-triazol-4-yl)-5-chlorophenyl)(phenyl)methanone) reported by the EU EWS in 2014 was not included as it was not available as a reference standard at the time of the study. In addition, three of the latest DBZDs reported by the EUDA in 2024 are open-ring prodrugs, so the prodrugs themselves were not tested, but their primary active metabolites were tested. For more information, see Section "All BZD metabolites and most prodrugs demonstrate GABA$_A$ activity".

Most of the (D)BZDs were found to be PAMs, as shown in Table 2, where the efficacies ($E_{max}$) ranged from 1.19 to 4.30 (Fig. 2C) and the potencies (EC$_{50}$) ranged from 0.363 to 150 nM (Fig. 2D). Deschloroetizolam (**43**), cloniprazepam (**9**), and desalkylgidazepam (**11**) were the most efficacious (4.30, 4.14, and 3.83, respectively), as shown in their concentration-response curves in Fig. 2A and the bar chart of the efficacies of all compounds in Fig. 2C. Clonazolam (**33**), delorazepam (**10**), and 2′-bromonordazepam (**2**) were the most potent with sub nanomolar potencies (0.363–0.862 nM), resulting in compounds 25–60 times more potent than diazepam (**13**), as shown in their concentration-response curves in Fig. 2B and the bar chart of the potencies of all compounds in Fig. 2D. There was no correlation found between the potency and efficacy and the year of notification to the EU EWS of the DBZDs. This data demonstrates that there is a large variability in the pharmacological properties of the emerging DBZDs and there is no trend suggesting clandestine producers are producing more potent or efficacious BZD.

Only five (D)BZDs showed little to no activity: rilmazafone (**53**), adinazolam (**29**), tofisopam (**50**), 2,4′-dichlorodiazepam (**3**; **Ro5-6900**), and 4′-chloro deschloroalprazolam (**28**; see Table 2). As rilmazafone (**53**) and adinazolam (**29**) are both prodrugs, they were not expected to have much activity, as discussed further in "All BZD metabolites and most prodrugs demonstrate GABA$_A$ activity". 2,4′-dichlorodiazepam (**3**), 4′-chloro deschloroalprazolam (**28**), and tofisopam (**50**) showed no significant activity, suggesting they are not PAMs of the GABA$_A$ receptor. 4′-chlorodiazepam (**5**; Ro 5-4864) and 4′-fluorodiazepam (**6**) were found to be negative allosteric modulators (NAMs) with $I_{min}$ of 0.159 and 0.333, respectively, and IC$_{50}$ of 189 and 1190 nM, respectively.

**Different base structures demonstrate equivalent GABA$_A$ activity.** Of the DBZDs reported to the EU EWS, the 1,4-BZD was the most prevalent base structure ($n = 15$), followed by the 1,3,4-triazolo ($n = 9$) and thienotriazolo ($n = 8$), with no correlation over time of the detection of DBZDs with different base structures. When looking at direct comparisons of close analogs with different base structures, there were no clear SARs identified for the base structures, as can be seen by the spread of efficacies and potencies for BZDs with different base structures in Fig. 2C, D, respectively. The concentration-response curves and statistical analysis for these comparisons (ANOVA, $\alpha = 0.05$) can be found in Supplementary Information Section 2 and Supplementary Table 1. There was also no significant difference between the potencies or efficacies of the different base structures ($p = 0.2$), indicating the 1,4-BZD, 1,3,4-triazolo, and thienotriazolo base structures have the same potential to activate the GABA$_A$ receptor.

**Substitutions on the benzene or thieno ring ($R_4$ or $R_5$ in Table 1) affect GABA$_A$ activity.** One of the most common substitutions to the BZD structure made by clandestine producers is at position 7 on the benzene ring of 1,4-BZDs and triazolos ($R_4$ in Table 1) or at position 2 on the thieno ring of thienotriazolos ($R_5$ in Table 1). A comparison of the potency and efficacy of analogs with different substitutions at these positions is shown in Fig. 3 with an overview of the statistical significance of the different comparisons. The structures of all the BZDs with the

**Table 2 | Efficacy ($E_{max}$ or $I_{min}$) and potency ($EC_{50}$ or $IC_{50}$) calculated for the benzodiazepines examined in this study, including the 95% confidence interval (CI), and the results of the antagonist competition studies, including the observed activity and the efficacy and potency of the resulting activity**

| Base | Name | Standard | | | | Antagonist competition | | | | |
|------|------|----------|--|--|--|------------------------|--|--|--|--|
| | | Efficacy | | Potency (nM) | | Observed activity | Efficacy | | Potency (nM) | |
| | | $E_{max}$ | 95% CI | $EC_{50}$ | 95% CI | | $E_{max}$ | 95% CI | $EC_{50}$ | 95% CI |
| A | (1) 2′-bromodiazepam | 2.99 | 2.79–3.19 | 3.67 | 2.33–5.72 | None | - | - | - | - |
| | (2) 2′-bromonordazepam | 2.39 | 2.24–2.55 | 0.862 | 0.435–1.54 | Potentiation | 1.65 | ND | ND | ND |
| | (3) 2,4′-dichlorodiazepam | 1.05[a] | ND | ND | ND | Inhibition[b] | $I_{min}$ = 0.752 | 0.607–0.843 | $IC_{50}$ = 158 | 28.0–977 |
| | (4) 3-hydroxyphenazepam | 3.78 | 3.16–4.49 | 3.79 | 1.24–10.4 | None | - | - | - | - |
| | (5) 4′-chlorodiazepam[b] | $I_{min}$ = 0.159 | 0.052–0.257 | $IC_{50}$ = 189 | 120–297 | Inhibition[b] | $I_{min}$ = 0.194 | 0.059–0.311 | $IC_{50}$ = 274 | 159–468 |
| | (6) 4′-fluorodiazepam[b] | $I_{min}$ = 0.333 | 0.201–0.425 | $IC_{50}$ = 1190 | 805–1850 | Inhibition[b] | $I_{min}$ = 0.00 | ?–0.114 | $IC_{50}$ = 3020 | 2390–3290 |
| | (7) Cinazepam | 3.34 | 3.04–3.69 | 148 | 83.9–253 | None | - | - | - | - |
| | (8) Clonazepam | 2.88 | 2.67–3.10 | 2.95 | 1.64–5.17 | - | - | - | - | - |
| | (9) Cloniprazepam | 4.14 | 3.52–4.82 | 26.3 | 7.21–74.9 | None | - | - | - | - |
| | (10) Delorazepam | 1.92 | 1.78–2.07 | 0.724 | 0.338–1.34 | None | - | - | - | - |
| | (11) Desalkylgidazepam | 3.83 | 3.33–4.41 | 13.9 | 4.94–35.1 | None | - | - | - | - |
| | (12) Desmethylflunitrazepam | 2.05 | 1.90–2.21 | 5.51 | 2.15–13.2 | None | - | - | - | - |
| | (13) Diazepam | 3.29 | 2.93–3.68 | 22.0 | 10.3–43.2 | None | - | - | - | - |
| | (14) Diclazepam | 2.96 | 2.71–3.22 | 1.48 | 0.775–2.62 | None | - | - | - | - |
| | (15) Difludiazepam | 2.13 | 1.86–2.33 | 1.50 | 0.370–4.21 | Potentiation | 1.47 | 1.26–1.89 | 363 | 49.2–1790 |
| | (16) Flubromazepam | 2.82 | 2.64–3.00 | 5.24 | 3.39–7.93 | None | - | - | - | - |
| | (17) Fludiazepam | 2.85 | 2.69–3.00 | 7.04 | 4.14–11.9 | - | - | - | - | - |
| | (18) Gidazepam | 1.94 | 1.67–2.25 | 13.2 | 1.69–48.1 | None | - | - | - | - |
| | (19) Lorazepam | 2.93 | 2.58–3.31 | 4.97 | 1.68–13.7 | - | - | - | - | - |
| | (20) Meclonazepam | 1.49 | 1.37–1.63 | 5.11 | 1.21–19.6 | None | - | - | - | - |
| | (21) Methylclonazepam | 1.99 | 1.92–2.06 | 3.51 | 2.36–5.15 | None | - | - | - | - |
| | (22) Nifoxipam | 1.87 | 1.61–2.20 | 93.1 | 22.2–372 | None | - | - | - | - |
| | (23) Nitrazepam | 1.59 | 1.44–1.79 | 51.9 | 19.9–137 | - | - | - | - | - |
| | (24) Nordazepam | 3.19 | 2.77–3.65 | 23.1 | 7.55–59.4 | None | - | - | - | - |
| | (25) Norfludiazepam | 1.61 | 1.53–1.69 | 4.33 | 1.84–9.75 | None | - | - | - | - |
| | (26) Phenazepam | 2.12 | 1.91–2.36 | 0.890 | 0.35–2.12 | None | - | - | - | - |
| | (27) Temazepam | 1.63 | 1.43–1.84 | 3.95 | 0.628–17.0 | - | - | - | - | - |
| B | (28) 4′–chloro deschloroalprazolam | 1.19[a] | ND | ND | ND | Inhibition[a] | $I_{min}$ = 0.611 | 0.479–0.727 | $IC_{50}$ = 17.8 | 4.69–78.3 |
| | (29) Adinazolam | 1.87[a] | ND | ND | ND | - | - | - | - | - |
| | (30) Alprazolam | 2.84 | 2.65–3.03 | 14.7 | 8.41–24.8 | - | - | - | - | - |
| | (31) Bromazolam | 2.45 | 2.22–2.70 | 15.1 | 6.33–34.1 | None | - | - | - | - |
| | (32) Clobromazolam | 3.18 | 2.83–3.55 | 3.50 | 1.61–7.03 | Potentiation | 1.89 | ND | ND | ND |
| | (33) Clonazolam | 1.54 | 1.40–1.70 | 0.363 | 0.082–1.24 | Potentiation | 1.43 | ND | ND | ND |
| | (34) Flualprazolam | 3.11 | 2.88–3.35 | 4.61 | 3.08–6.75 | None | - | - | - | - |
| | (35) Flubromazolam | 3.03 | 2.66–3.44 | 1.55 | 0.657–3.29 | - | - | - | - | - |
| | (36) Flunitrazolam | 2.64 | 2.39–2.91 | 8.40 | 3.01–21.9 | None | - | - | - | - |
| | (37) Nitrazolam | 1.43 | 1.30–1.57 | 6.25 | 5.34–42.7 | None | - | - | - | - |
| | (38) Pyrazolam | 1.61 | 1.51–1.72 | 13.3 | 5.00–33.0 | None | - | - | - | - |
| C | (39) Rilmazolam | 1.82 | 1.66–1.99 | 12.4 | 4.88–38.4 | None | - | - | - | - |
| D | (40) Brotizolam | 3.62 | 3.31–3.94 | 1.87 | 0.92–3.62 | None | - | - | - | - |
| | (41) Clotizolam | 3.15 | 2.86–3.45 | 6.47 | 3.50–11.8 | - | - | - | - | - |
| | (42) Deschloroclotizolam | 2.35 | 2.18–2.52 | 65.1 | 37.4–111 | Potentiation | 1.40 | 1.23–2.06 | 21.1 | 2.60–215 |
| | (43) Deschloroetizolam | 4.30 | 3.62–5.06 | 54.6 | 17.8–148 | None | - | - | - | - |
| | (44) Etizolam | 3.19 | 2.83–3.59 | 12.0 | 5.09–26.7 | None | - | - | - | - |
| | (45) Flubrotizolam | 2.68 | 2.43–2.94 | 3.34 | 1.59–6.51 | None | - | - | - | - |
| | (46) Fluclotizolam | 2.62 | 2.43–2.81 | 5.87 | 3.28–10.2 | None | - | - | - | - |
| | (47) Fluetizolam | 3.66 | 3.25–4.14 | 150 | 72.9–296 | None | - | - | - | - |
| | (48) Metizolam | 2.01 | 1.75–2.34 | 49.9 | 19.1–130 | None | - | - | - | - |

**Table 2 (continued) | Efficacy ($E_{max}$ or $I_{min}$) and potency (EC$_{50}$ or IC$_{50}$) calculated for the benzodiazepines examined in this study, including the 95% confidence interval (CI), and the results of the antagonist competition studies, including the observed activity and the efficacy and potency of the resulting activity**

| Base | Name | Standard | | | | Antagonist competition | | | | |
| --- | --- | --- | --- | --- | --- | --- | --- | --- | --- | --- |
| | | Efficacy | | Potency (nM) | | Observed activity | Efficacy | | Potency (nM) | |
| | | $E_{max}$ | 95% CI | EC$_{50}$ | 95% CI | | $E_{max}$ | 95% CI | EC$_{50}$ | 95% CI |
| E | **(49)** Thionordiazepam | 1.76 | 1.61–1.92 | 51.4 | 18.5–136 | Potentiation | 1.26 | ND | ND | ND |
| F | **(50)** Tofisopam | 1.08[a] | ND | ND | ND | None | - | - | - | - |
| G | **(51)** Bretazenil | 1.39 | 1.31–1.47 | 2.38 | 0.955–5.44 | None | - | - | - | - |
| H | **(52)** Bentazepam | 2.68 | 2.38–3.03 | 92.7 | 39.8–216 | None | - | - | - | - |
| I | **(53)** Rilmazafone | 1.38[a] | ND | ND | ND | - | - | - | - | - |
| J | **(54)** Flumazenil[c] | **$I_{min}$ = 1.254** | **0.952–1.51** | **IC$_{50}$ = 0.798** | **0.196–3.98** | - | - | - | - | - |

For the antagonist competition studies, the observed activity is listed as "None" for benzodiazepines that were observed to have a complete reduction in activity in the presence of flumazenil.

*ND* not determined (values could not be calculated as saturation was not reached).

Bold: Benzodiazepines and values in bold showed inhibition rather than potentiation, so efficacy is $I_{min}$ values and potency is IC$_{50}$ values.

[a]Greatest efficacy achieved with a concentration between 2.4 and 3000 nM.

[b]Negative allosteric modulator, so efficacy is $I_{min}$ values, and potency is IC$_{50}$ values.

[c]Benzodiazepine antagonist. Curve is based on competitive antagonism of diazepam at EC$_{80}$ concentration.

concentration-response curves and data from the statistical analysis for these comparisons can be found in Supplementary Fig. 5, Supplementary Section 4, and Supplementary Table 2.

For 1,4-BZDs, analogs with a chlorine, bromine, or nitro group on the fused benzene ring ($R_4$ in Table 1) were equally potent, except delorazepam **(10)** with a chlorine was significantly more potent than clonazepam **(8)** with a nitro group ($p = 0.01$). On the other hand, analogs with a chlorine were the least efficacious, with efficacies 1.1-1.75 times lower than analogs with a bromine or nitro group. The efficacy relationship between analogs with a bromine or nitro group is unclear, as flubromazepam **(16;** Br) was 1.4 times more efficacious ($p < 0.01$) than desmethylflunitrazepam **(12;** NO$_2$), but clonazepam **(8;** NO$_2$) was 1.4 times more efficacious ($p < 0.01$) than phenazepam **(26;** Br).

For the 1,3,4-triazolos, there are no clear SARs for the potency, as each of the three structurally related groups have different relationships. For example, clonzolam **(33)** with a nitro is significantly more potent than clobromazolam **(32)** with a bromine ($p < 0.05$), but flubromazolam **(35)** with a bromine is significantly more potent than flunitrazolam **(36)** with a nitro ($p < 0.05$), as can be seen in Fig. 3. For efficacy, the chlorine analogs were equally as efficacious as the bromine analogs, but 1.2 times more efficacious than nitro analogs ($p = 0.04$) based on one case. The bromine analogs were either equally as efficacious (flubromazolam **(35;** Br) ≈ flunitrazolam **(36;** NO$_2$), where $p = 0.3$) or more efficacious than the nitro group analogs (clobromazolam **(32;** Br) was 2.1 times more efficacious than clonazolam **(33;** NO$_2$), where $p < 0.01$).

For the thienotriazolos, analogs with a bromine on the thieno ring ($R_5$ in Table 1) were the most potent (3.5 times more potent than chlorine analogs and 6.4–45 times more potent than ethyl analogs) followed by chlorine analogs (1.8 times more potent than ethyl analogs) and the ethyl group, where the majority of these relationships were statistically significant as shown in Fig. 3. The relationships for efficacy are unclear as brotizolam **(40;** Br) was more efficacious than etizolam **(44;** ethyl), while the opposite was true for flubrotizolam **(45;** Br) and fluetizolam **(47;** ethyl). In addition, the efficacy of etizolam **(44;** ethyl) was almost the same as clotizolam **(41;** Cl) (3.19 and 3.15, respectively).

**Halogenation at the 2, 4, and 6 positions on the phenyl ring ($R_{6-8}$ in Table 1) affects potency.** Another common substitution to the BZD structure made by clandestine producers is at position 2 on the phenyl ring ($R_6$), but DBZDs with substitutions at the 4 ($R_7$) and 6 ($R_8$) positions have also emerged on the recreational market. A comparison of the potency of analogs with different substitutions at these positions is shown in Fig. 4 with an overview of the statistical significance of the different comparisons. The structures of all the BZDs with the concentration-

response curves and data from the statistical analysis for these comparisons can be found in Supplementary Information Section 5, Supplementary Fig. 6, and Supplementary Tables 3–5.

For 1,4-BZDs, a chlorine at the 2 position was 15-32 times more potent than a hydrogen ($p < 0.01$) and 4.8-6.0 times more potent than a fluorine, except for clonazepam **(8)** with a chlorine, which was equally as potent as desmethylflunitrazepam **(12)** with a fluorine ($p = 0.5$). A fluorine was equally as potent as a hydrogen, except in the case of nitrazepam **(23)** with a hydrogen which was 9.4-fold less potent than desmethylflunitrazepam **(12;** $p = 0.03$). Examination of the prophetic 2'-bromo analogs (2'-bromodiazepam **(1)** and 2'-bromonordazepam **(2)**) found that a bromine at the 2 position was 6.0- to 27-fold more potent than a hydrogen ($p < 0.01$) but equally as potent as a fluorine and chlorine ($p > 0.05$). Each of the four available structural comparisons shows a different relationship for efficacy, so the relationships for efficacy for 1,4-BZDs are unclear, although both of the 2'-bromo analogs were equally as efficacious as the hydrogen analogs. There was also one 1,4-BZD that had a fluorine at both the 2 ($R_6$) and 6 ($R_8$) positions (difludiazepam **(15)**), which was found to be 4.7 times more potent ($p = 0.05$) and 1.3 times less efficacious ($p < 0.01$) than fludiazepam **(17)** with only one fluorine at the 2 position.

For 1,3,4-triazolos, a chlorine at the 2 position was 4.3-17.2 times more potent than a hydrogen, although the differences were not found to be statistically significant. There were two structural comparisons where a fluorine was also 3.2–9.8 times more potent than a hydrogen ($p < 0.04$), except nitrazolam **(37)** with a hydrogen was equally as potent as flunitrazolam **(36)** with a fluorine ($p = 1.0$). The relationship between a fluorine and chlorine was less clear as flubromazolam **(35)** with a fluorine was equally as potent as clobromazolam **(32)** with a chlorine ($p = 0.4$), while clonazolam **(33)** with a chlorine was 23 times more potent than flunitrazolam **(36)** with a fluorine ($p < 0.01$). For efficacy, fluorine and chlorine were 1.1-1.8 times more efficacious than the hydrogen in all cases, but the difference was only statistically significant in half of the cases. It should also be noted that pyrazolam **(38)**, which has a nitrogen on the ring at the 2 position, was equally as potent ($p < 1$) and significantly less efficacious ($p < 0.01$) than bromazolam **(31)** with a carbon on the ring.

For thienotriazolos, the SARs for substitutions at the 2 position vary between the three available structural comparisons, so the SARs for both potency and efficacy are unclear. For example, deschloroetizolam **(43)** with a hydrogen is not significantly different in potency from fluetizolam **(47)** with a fluorine ($p = 0.3$), but fluclotizolam **(46)** with a fluorine is 11 times more potent than deschloroclotizolam **(42)** with a hydrogen ($p < 0.01$).

The para (4 position; $R_7$) halogenated analogs of diazepam **(13)**, 4'-fluorodiazepam **(6)** and 4'-chlorodiazepam **(5)**, were both NAMs,

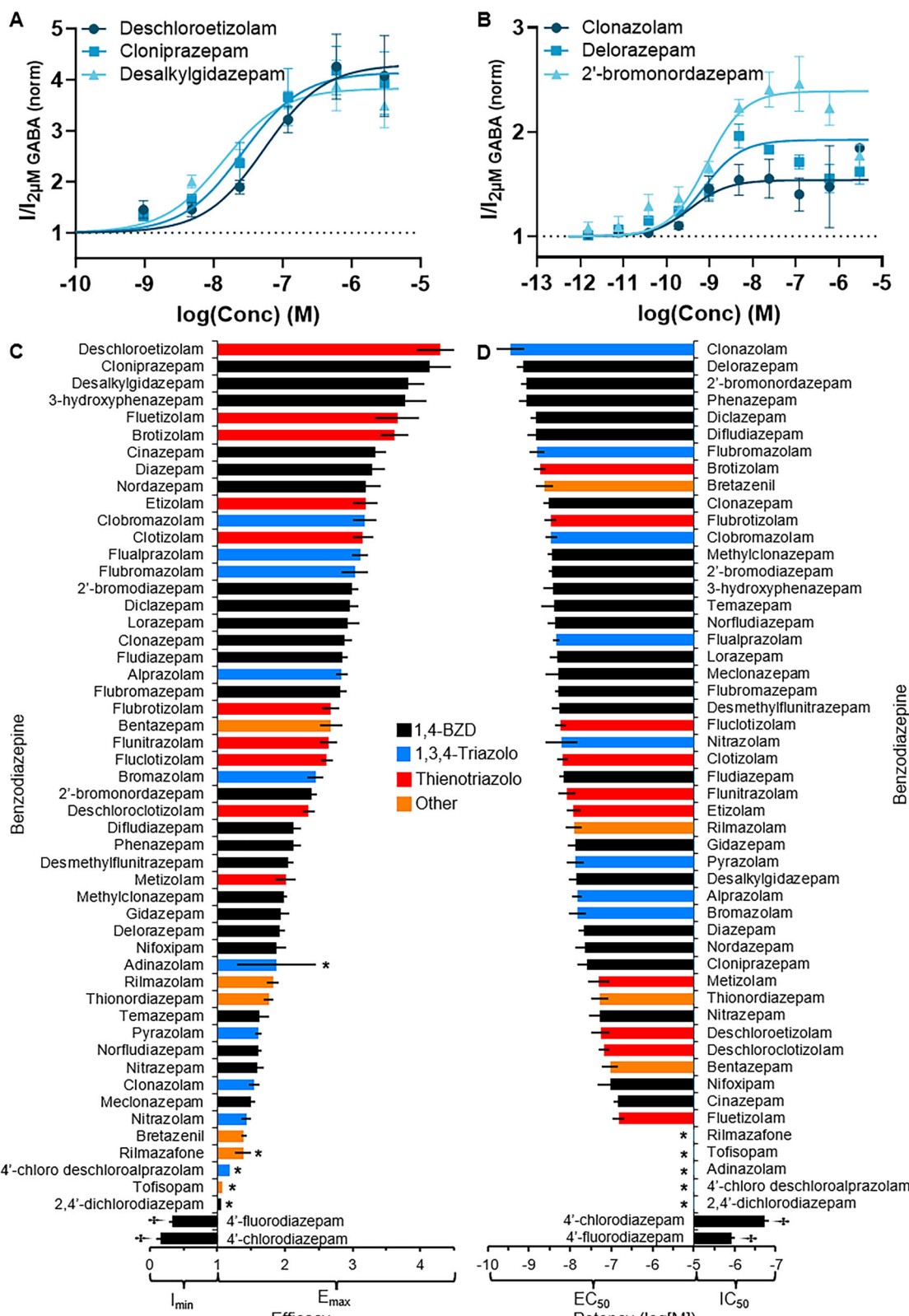

**Fig. 2 | Overview of the potency and efficacy of all benzodiazepines (BZDs) examined in this study.** Concentration-response curves of the three most **A** efficacious and **B** potent BZDs included in this study. The data was normalized so the baseline GABA (2 μM) activity was at an efficacy of 1, which is shown as a dotted line. **C** The efficacy and **D** potency of all BZDs examined in this study organized from most to least efficacious/potent, where the axis of **C** is set to the baseline GABA activity of 1. The colors of the bars indicate the base group of the compound. Data for each compound is from a minimum of four independent experiments ($n \geq 4$). Error bars indicate standard error to the mean (SEM). † indicates compounds that are negative allosteric modulators (NAMs), so the $I_{min}$ is reported for efficacy, and $IC_{50}$ reported for potency. * indicates BZDs that did not reach saturation, so the efficacy reported is the greatest efficacy achieved with a concentration between 2.4 and 3000 nM, and the potency was not able to be calculated.

**Fig. 3 | Relationship between GABA$_A$ receptor potency (EC$_{50}$) and efficacy ($E_{max}$) and substitutions at position 7 on the fused benzene ring ($R_4$) or 2 position on the fused thieno ring ($R_5$) of benzodiazepines (BZDs).** Comparison of the EC$_{50}$ and $E_{max}$ of substitutions at position 7 on the fused benzene ring ($R_4$) of (**A, D**) 1,4-BZDs and (**B, E**) 1,3,4-triazoloBZDs and substitutions at the 2 position on the fused thieno ring ($R_5$) of (**C, F**) thieno-triazoloBZDs. Data for each compound is from a minimum of four independent experiments ($n \geq 4$). Error bars indicate standard error to the mean (SEM). The results of statistical comparisons from Brown-Forsythe and Welch ANOVA tests ($\alpha = 0.05$) between BZD analogs with different substitutions are provided where * indicates statistically significant and "ns" indicates not statistically significant. Complete statistical data and concentration-response curves can be found in the Supplementary Information Sections 3–4 and Supplementary Table 2.

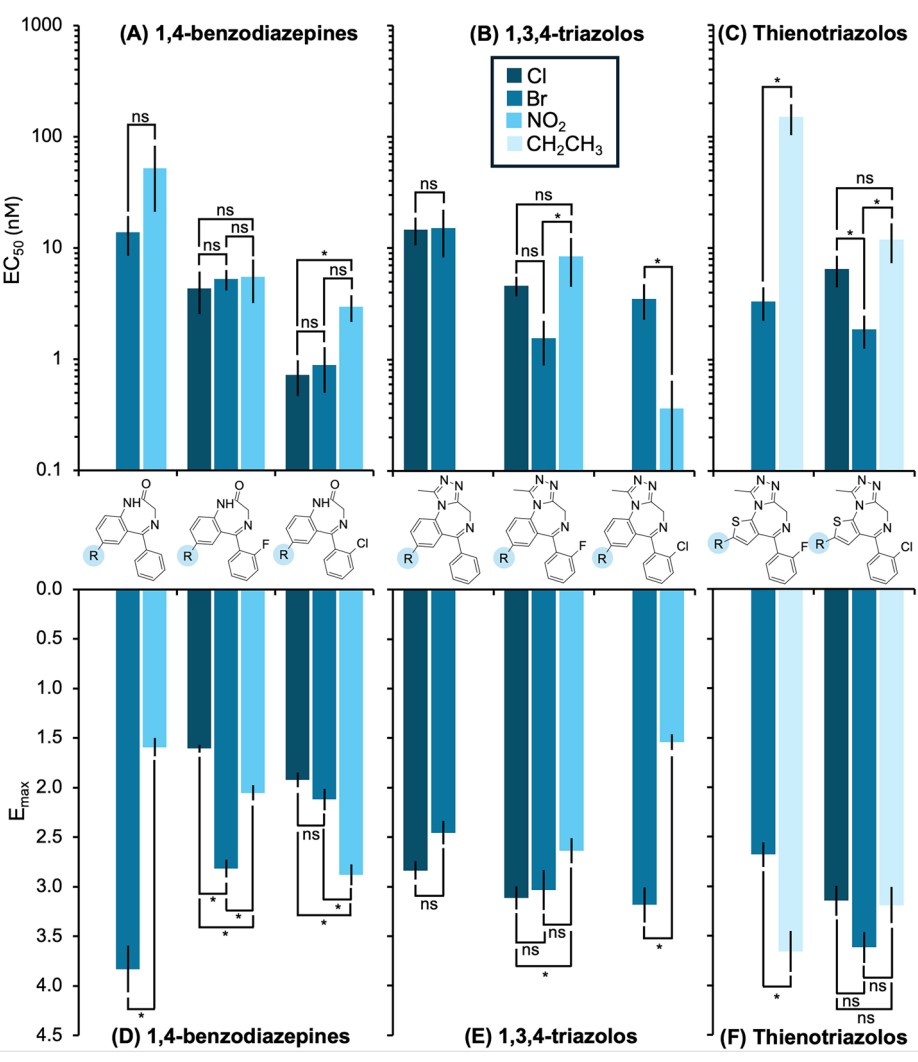

leading to inhibition rather than potentiation of the GABA$_A$ receptor. 4′-chlorodiazepam (**5**) was 6.3 times more potent ($p < 0.01$) than 4′-fluorodiazepam (**6**), but equally efficacious ($p = 0.1$). This is similar to the SARs observed for halogenation at the 2 position of the phenyl ring. On the other hand, there was little to no activity observed for 2,4′-dichlorodiazepam (**3**) and 4′-chloro deschloroalprazolam (**28**). Overall, this indicates that the position of halogenation on the phenyl ring is crucial for the type of activity displayed by BZDs, where a total reversal of the activity from positive to negative allosteric modulation or complete reduction of activity is observed when moving the halogen from the 2 to 4 position.

**All BZD metabolites and most prodrugs demonstrate GABA$_A$ activity.** Figure 5 shows comparisons of the GABA$_A$ activity of prodrugs and their primary metabolite and parent compounds to one or more of their metabolites. Metabolism data are not available for some DBZDs, but their likely metabolites are included, such as clonazepam (**8**) as the likely demethylated metabolite of methylclonazepam (**21**).

All the prodrugs (cinazepam (**7**), cloniprazepam (**9**), and gidazepam (**18**)) showed some activity, except rilmazafone (**53**) for which an EC$_{50}$ value was unable to be calculated due to it never reaching saturation. All the investigated metabolites also showed activity, including the demethylated (nor) metabolites, which lack a methyl group at the 1 position on the diazepine ring ($R_1$) in comparison to their parent, and the 3-hydroxy metabolites, which have a hydroxy group added at the 3 position on the diazepine ring ($R_3$) in comparison to the parent. Three of the demethylated metabolites (delorazepam (**10**), norfludiazepam (**25**), and

2′-bromonordazepam (**2**)) were 1.3- to 1.8-fold less efficacious ($p < 0.01$) than their parent (diclazepam (**14**), fludiazepam (**17**), and 2′-bromodiazepam (**1**), respectively), while nordazepam (**24**) was equally efficacious as its parent diazepam (**13**) and clonazepam (**8**) was 1.5-fold more efficacious than its likely parent methylclonazepam (**21**; $p < 0.01$). 2′-bromonordazepam (**2**) was 4.3-fold more potent than its parent, while the rest of the demethylated metabolites were equally as potent as the parent. This demonstrates that the addition of a methyl group at the 1 position on the diazepine ring ($R_1$) generally reduces efficacy but does not change the potency of the compound.

Examining the prodrugs cloniprazepam (**9**) and gidazepam (**18**), which have bulkier moieties at $R_1$ (methylcyclopropyl and acetohydrazide, respectively), cloniprazepam (**9**) was 1.4-fold more efficacious ($p = 0.04$) and 8.9-fold less potent ($p < 0.04$) than its primary metabolite clonazepam (**8**), while gidazepam (**18**) was 2.0-fold less efficacious ($p = 0.01$) and equally as potent as its primary metabolite desalkylgidazepam (**11**). Therefore, while it is not clear if the presence of a bulkier moiety at $R_1$ reduces the potency or efficacy, it seems it will reduce the GABA$_A$ activity by significantly reducing at least one of them. The addition of a bulkier moiety at the 2 position of the thieno ring ($R_2$) for 1,3,4-triazolos also significantly reduced the GABA$_A$ activity. This can be seen with the prodrug adinazolam (**29**) with a $N,N$-dimethylmethanamine at $R_2$, which did not reach saturation, in comparison to its close analog alprazolam (**30**) with a methyl at $R_2$, which has an EC$_{50}$ of 14.7 nM and a 1.5-fold greater $E_{max}$ than the greatest efficacy achieved for adinazolam (**29**). Adinazolam (**29**) is not included in Fig. 5 as its primary metabolite, $N$-desmethyl-adinazolam, was not examined in this study.

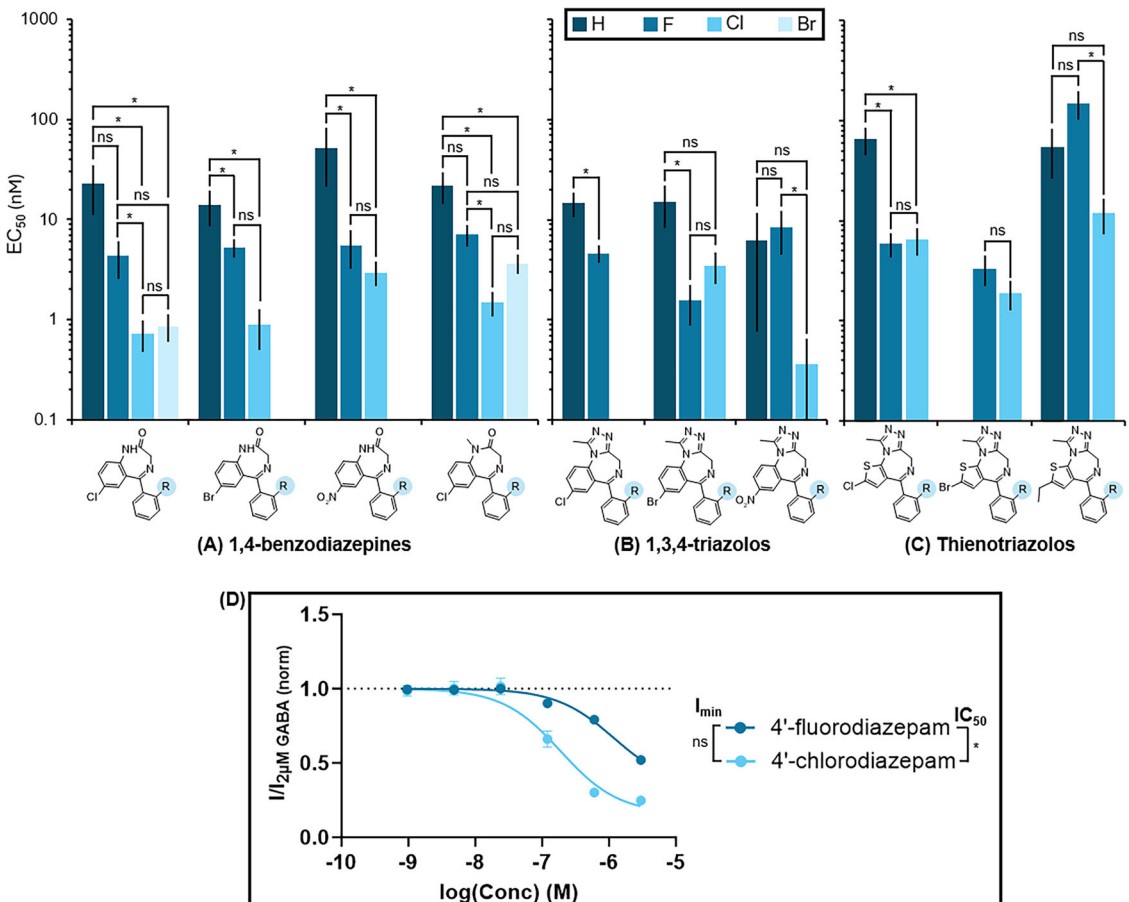

**Fig. 4 | Relationship between GABA$_A$ receptor potency (EC$_{50}$) and halogenation at the 2 ($R_6$) and 4 ($R_7$) position on the phenyl ring of benzodiazepines (BZDs).** Comparison of the EC$_{50}$ of halogenation at the 2 position on the phenyl ring ($R_6$) of **A** 1,4-BZDs, **B** 1,3,4-triazoloBZDs, and **C** thienotriazoloBZDs. **D** Comparison of the in vitro GABA$_A$ receptor activity of halogenation at the 4 position on the phenyl ring ($R_7$) of 1,4-BZDs. Data for each compound is from a minimum of four independent experiments ($n \geq 4$). Error bars indicate standard error to the mean (SEM). The results of statistical comparisons from Brown-Forsythe and Welch ANOVA tests ($\alpha = 0.05$) between the different halogen analogs for each BZD structural group are provided, where * indicates statistically significant and "ns" indicates not statistically significant. Complete statistical data and concentration-response curves can be found in Supplementary Information Section 5, Supplementary Fig. 6, and Supplementary Tables 3–5.

Looking at the 3-hydroxy metabolites, temazepam (**27**) was equally as potent as its parent diazepam (**13**), while the rest (3-hydroxyphenazepam (**4**), lorazepam (**19**), and nifoxipam (**22**)) were 4.3- to 17-fold less potent than their parents (phenazepam (**26**), delorazepam (**10**), and desmethylflunitrazepam (**12**), respectively), although this difference was only statistically significant for lorazepam (**19**; $p = 0.03$). This demonstrates that the addition of a hydroxy group at the 3 position on the diazepine ring ($R_3$) generally reduces the potency, except in the case of temazepam (**27**) and its parent diazepam (**13**). However, there was no clear SAR for efficacy as lorazepam (**19**) and 3-hydroxyphenazepam (**4**) were 1.5- and 1.8-fold more efficacious ($p = 0.01$), respectively, than their parents, while nifoxipam (**22**) was equally as efficacious as its parent and temazepam (**27**) was 2.0-fold less efficacious than its parent ($p < 0.01$). The prodrug cinazepam (**7**) with a bulky moiety (butanedioic acid) at $R_3$ was 39-fold less potent ($p < 0.01$) and equally as efficacious as its metabolite 3-hydroxyphenazepam (**4**; $p = 0.6$). Although there is only one comparison, this data suggests that a bulky moiety at $R_3$ significantly reduces potency, but more BZDs with a bulky moiety at $R_3$ should be examined to further elucidate this SAR.

**Some BZDs have remaining activity in the presence of the BZD antagonist flumazenil**
To determine if the $\alpha^+/\gamma_2^-$ interface (BZD binding site) is the main binding site for DBZDs, the same automated planar patch-clamp technique was used, but the BZD antagonist, flumazenil (**54**), was added to try to block the

BZD binding site. All DBZDs were screened in the presence of flumazenil (**54**) (600 or 3000 nM) by testing two concentrations of each DBZD, one around the EC$_{50}$ and one around the EC$_{80}$. See "Methods" for more information. For those that showed some activity in the presence of flumazenil (**54**), a full concentration-response curve was determined with 3000 nM flumazenil (**54**) and increasing concentrations of the compound of interest (0.96–3000 nM). As shown in Table 2, of the 42 DBZDs tested, 32 DBZDs plus diazepam showed a complete reduction in activity in the presence of flumazenil (**54**). An example concentration-response curve of a BZD with a complete reduction in activity is shown in Fig. 6A. This demonstrates that the observed activity of these DBZDs is largely mediated through the BZD interface.

The remaining DBZDs still showed some activity in the presence of flumazenil (**54**). Six DBZDs (2'-bromonordazepam (**2**), clobromazolam (**32**), clonazolam (**33**), deschloroclotizolam (**42**), difludiazepam (**15**), and thionordazepam (**49**)) showed some remaining potentiation, as can be seen in Fig. 6B–F. Concentration-response curves were able to be calculated for the remaining potentiation of difludiazepam (**15**) and deschloroclotizolam (**42**), but the others did not reach saturation, so curves could not be calculated; therefore, only the greatest observed efficacy is reported for those compounds. The efficacy of the potentiation in the presence of flumazenil (**54**) was 1.1–1.9 times lower than the efficacy of the compound's potentiation without flumazenil (**54**). The potency of difludiazepam (**15**) was reduced 240-fold in the presence of flumazenil (**54**), while it was increased

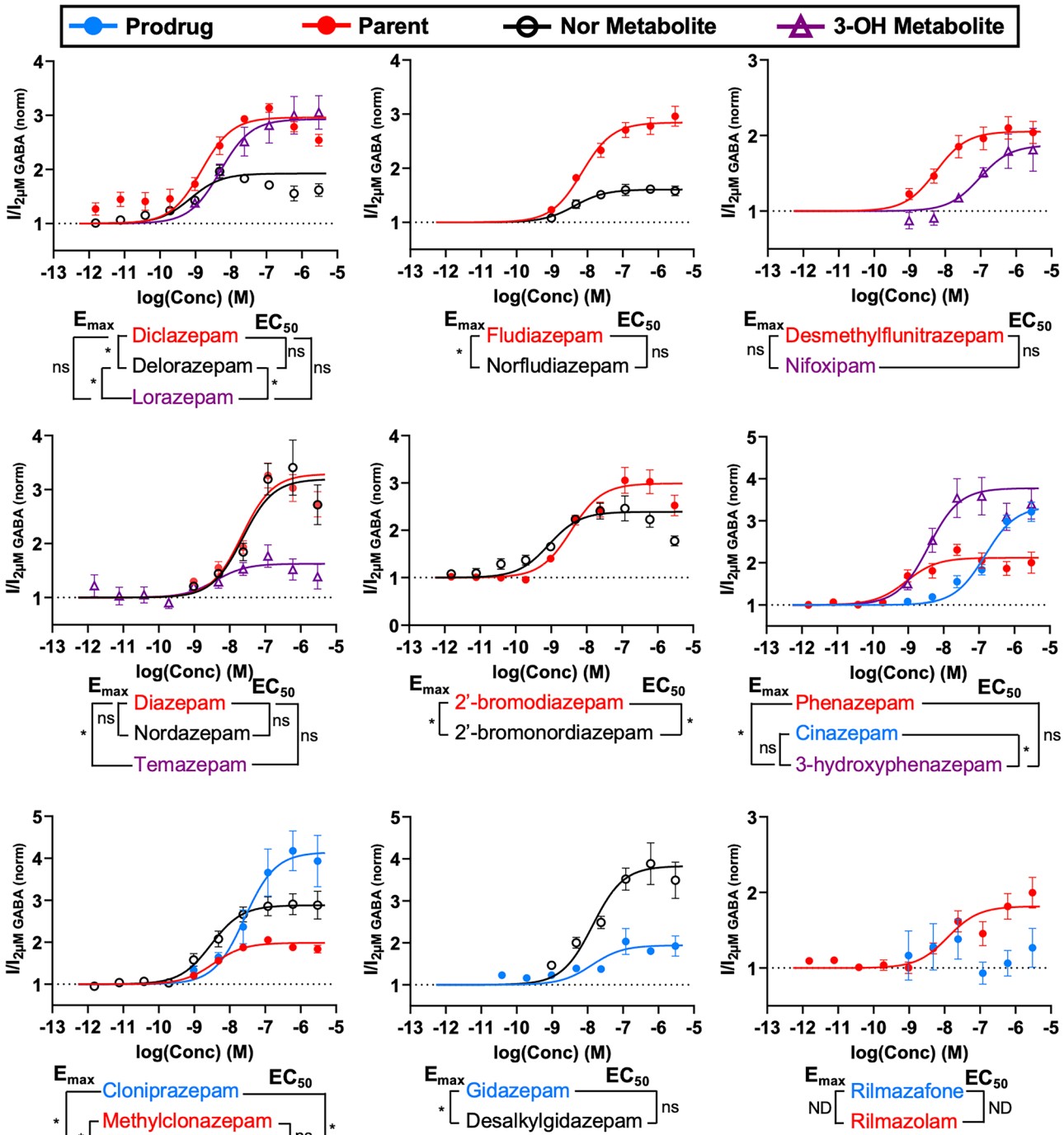

**Fig. 5 | Comparison of the in vitro GABA$_A$ receptor activity between parent compounds and their demethylated (nor) and/or 3-hydroxy metabolites and prodrugs and their primary metabolite.** Metabolism data for some designer benzodiazepines (DBZDs) are not available, but likely metabolites are included. The data were normalized so the baseline GABA$_A$ (2 μM) activity was at an efficacy of 1, which is shown as a dotted line. Data for each compound is from a minimum of four independent experiments ($n \geq 4$). Error bars indicate standard error to the mean (SEM). The results of statistical comparisons from Brown-Forsythe and Welch ANOVA tests ($\alpha = 0.05$) for the efficacy ($E_{max}$) and potency ($EC_{50}$) between the parent/prodrug and metabolite are provided, where * indicates statistically significant, "ns" indicates not statistically significant, and "ND" indicates not determined as a concentration-response curve was not able to be determined for a compound. Complete statistical data can be found in Supplementary Table 6.

10-fold for deschloroclotizolam (**42**). This indicates that these compounds may compete away flumazenil (**54**), but it is also possible the remaining potentiation is mediated though an interface on the GABA$_A$ receptor other than the BZD interface.

In the presence of flumazenil (**54**), the NAMs 4′-fluorodiazepam (**6**; Fig. 6G) and 4′-chlorodiazepam (**5**; Fig. 6H) showed the same inhibition as without flumazenil (**54**), where there was no significant difference between

the efficacy and potency of each DBZD with and without flumazenil (**54**) ($p > 0.1$; see Supplementary Table 7). This demonstrates that the inhibitory activity of these DBZDs results from interaction with a binding interface other than the BZD interface. 2,4′-dichlorodiazepam (**3**) and 4′-chloro deschloroalprazolam (**28**), which were found to have no activity, showed inhibition in the presence of flumazenil (**54**) (Fig. 6I). This suggests that they produce potentiation from the BZD interface and inhibition from another

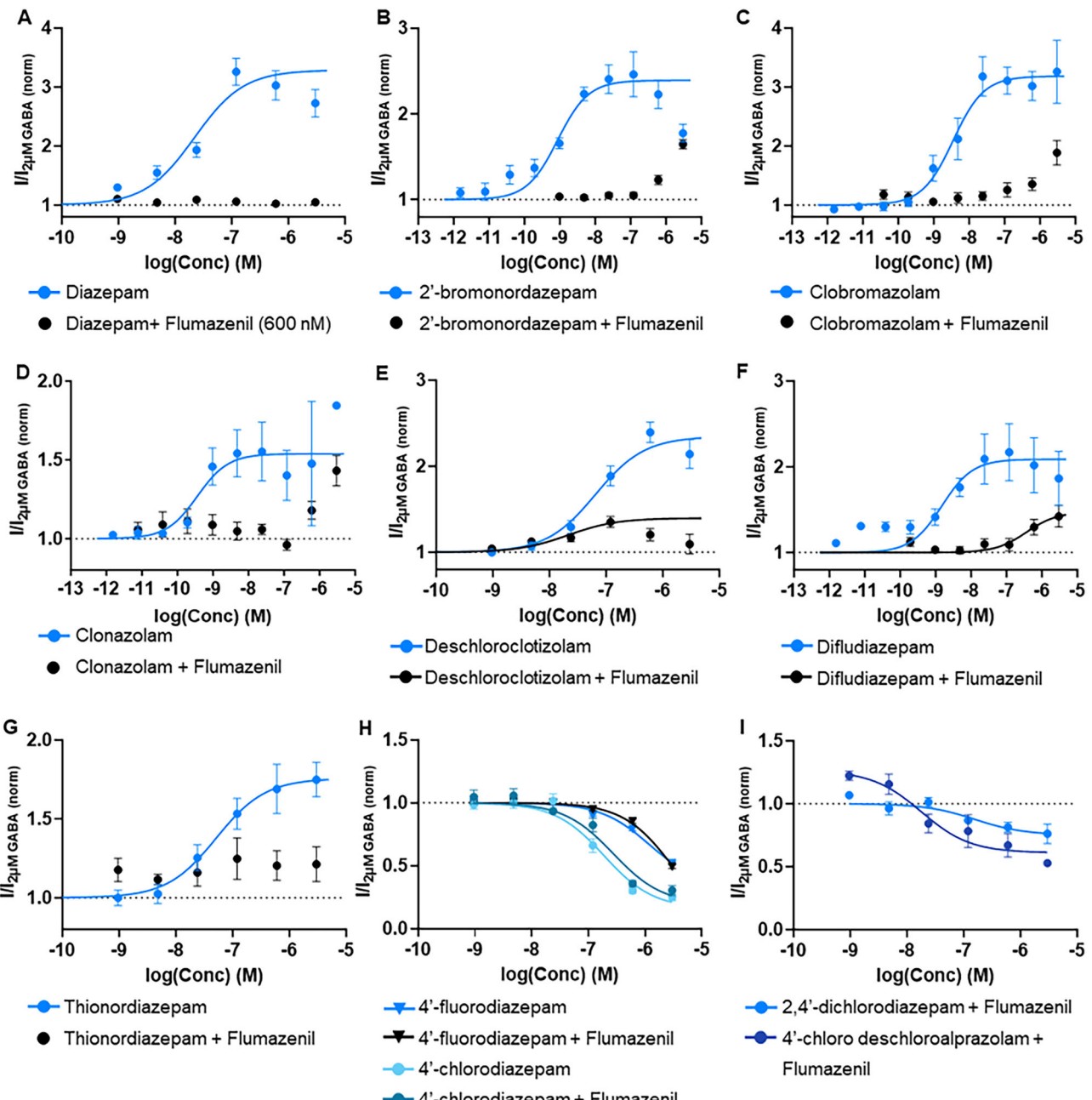

**Fig. 6 | Results from the antagonism experiments, where flumazenil (54) was added with increasing concentrations of the (designer) benzodiazepine ((D) BZD), in comparison to the standard concentration-response curves for the (D) BZD.** The data was normalized so the baseline GABA$_A$ (2 μM) or GABA$_A$ (2 μM) + Flumazenil activity was at an efficacy of 1, which is shown as a dotted line. Data for each compound is from a minimum of four independent experiments ($n \geq 4$). Error bars indicate standard error to the mean (SEM). **A** Example of a BZD (diazepam (**13**)) that showed a complete reduction of activity in the presence of flumazenil (**54**; 600 nM). **B–F** DBZDs with remaining potentiation in the presence of flumazenil (**54**); **C** DBZDs with inhibition with flumazenil (**54**) added (3000 nM). **G–I** DBZDs with inhibition in the presence of flumazenil (**54**; 3000 nM).

binding interface, resulting in the appearance of no activity without flumazenil (**54**).

## Discussion

In this study, the in vitro α$_1$β$_2$γ$_2$ GABA$_A$ receptor activity determined for 42 DBZDs and 11 prescription BZDs demonstrated a wide variability in the potency and efficacy of DBZDs released on the recreational market over time (Fig. 2C, D) with some (D)BZDs up to 60 times more potent than diazepam (**13**; Fig. 2A). While no studies are available on the relative efficacy and potency of DBZDs, there is anecdotal information from user forums and "trip reports" on suggested or reported dosages[30]. The differences between the suggested doses of available (D)BZDs relative to diazepam (**13**; see Supplementary Table 8) are mostly consistent with the differences between the potencies in this study, although the size of the differences may vary. For example, alprazolam (**30**) was 1.5-fold more potent than diazepam (**13**), but the suggested alprazolam (**30**) dose was 20-fold less than diazepam (**13**)[30]. This result isn't surprising as it is well known that in vitro potency can vary from in vivo therapeutic drug exposure (dosing)[31] due to the large number of variables affecting in vivo efficacious exposure levels, including bioavailability, clearance, route of administration, and duration of action[32]. It may also be due to BZDs interacting with other targets, including other GABA$_A$ isoforms[33] and mitochondrial translocator protein 18 kDa

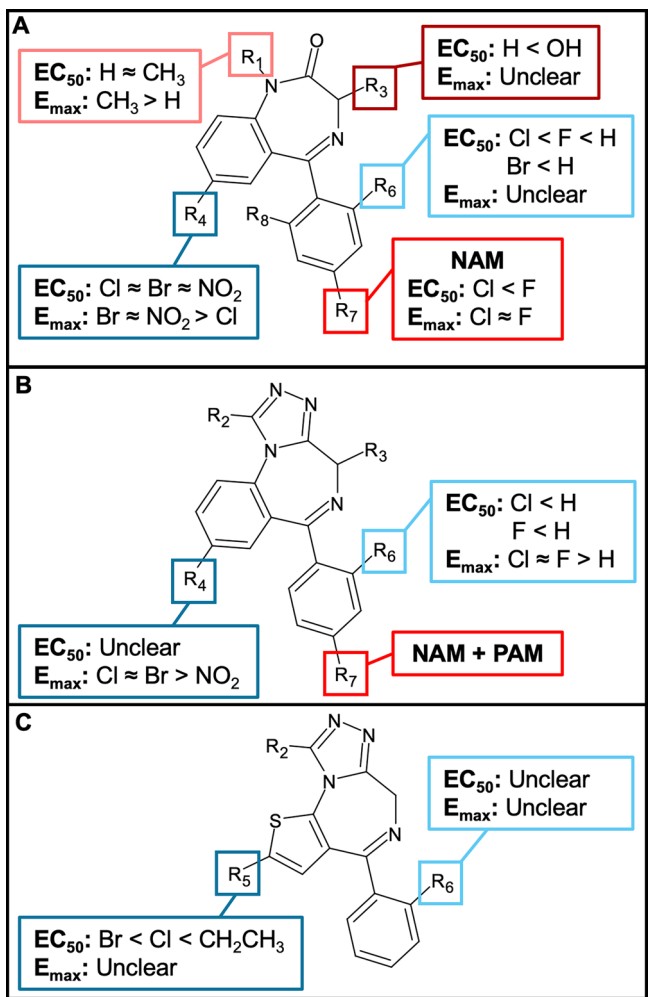

**Fig. 7 | Overview of the structure-activity relationships (SARs) for potency (EC$_{50}$) and efficacy ($E_{max}$) identified for benzodiazepines with different base structures.** The observed SARs discussed in detail in Sections "Different base structures demonstrate equivalent GABA$_A$ activity" to "All BZD metabolites and most prodrugs demonstrate GABA$_A$ activity" are summarized for **A** substitutions of 1,4-benzodiazepines at $R_1$, $R_3$, $R_4$, $R_6$, and $R_7$ (as shown in Table 1); **B** substitutions of 1,3,4-triazolobenzodiazepines at $R_4$, $R_6$, and $R_7$; and **C** substitutions of thieno-triazolobenzodiazepines at $R_5$ and $R_6$. If there was no clear SAR observed for any substitutions at a certain position, it is listed as "unclear".

(TSPO)[34], which were outside of the scope of this study to examine. However, the observed consistency of the in vitro $\alpha_1\beta_2\gamma_2$ GABA$_A$ potency with the relative dosing demonstrates that the in vitro $\alpha_1\beta_2\gamma_2$ GABA$_A$ potency can be useful for harm reduction advice about the potential strength of different DBZDs to inform cautious dosing.

Examination of 53 (D)BZDs in this study allowed for exploration of SARs, where significant SARs were identified for substitutions at position 7 on the benzene ring ($R_4$) for 1,4-BZDs and 1,3,4-triazolos; position 2 on the thieno ring of thienotriazolos ($R_5$); and positions 2, 4, and 6 on the phenyl ring, as summarized in Fig. 7. There is not any other experimental data with which to compare the SARs identified in this study, but it is possible to compare them to SARs from previously predicted binding affinities (log(1/c) values) calculated using quantitative structure-activity relationship (QSAR) modelling[35,36]. The QSAR binding affinity relationships were compared to the potency relationships from this study, as there was little agreement between the efficacy relationships in this study.

For substitutions at position 7 on the benzene ring ($R_4$) of 1,4-BZDs, all substitutions were equally potent, while a chlorine was the least efficacious. On the other hand, NO$_2$ was the least efficacious for 1,3,4-triazolos, and the

potency relationships were unclear. This is inconsistent with the SARs from QSAR modelling, as a nitro group was found to have the greatest affinity, followed by the chlorine and bromine[35,36]. In comparison, at the 2 position on the thieno ring ($R_5$) of thienotriazolos, a bromine was the most potent, followed by chlorine and ethyl, which is consistent with the QSAR modelling, where a bromine had greater binding affinity than an ethyl[35,36]. There was no data for clotizolam (**41**) from the QSAR modelling, so the SARs with the chlorine substitution could not be compared.

At position 2 on the phenyl ring ($R_6$), a chlorine was the most potent, followed by a fluorine and hydrogen for both 1,4-BZDs and 1,3,4-triazolos, while the potency relationships for thienotriazolos were unclear. The efficacy relationships here were unclear, except that the hydrogen was the least efficacious for the 1,3,4-triazolos. The hydrogen also showed the lowest activity for the binding affinity in the QSAR modelling, but the other SARs were inconsistent[35,36]. Finally, at position 4 on the phenyl ring ($R_7$), which produced NAMs, a chlorine was also found to be more potent than a fluorine, but equally efficacious.

By examining the relationships between prodrugs and parents with their demethylated (nor) and 3-hydroxy metabolites, it is possible to see that there are significant SARs at the 1 and 3 positions on the diazepine ring ($R_1$ and $R_3$, respectively) of 1,4-BZDs. At the 1 position ($R_1$), the parent compounds with a methyl were equally potent but more efficacious than their demethylated metabolites with a hydrogen. However, examination of prodrugs shows that the addition of a bulky group at the $R_1$, such as a methylcyclopropyl (e.g., cloniprazepam (**9**)) or acetohydrazide (e.g., gidazepam (**18**)), results in a significant reduction in activity, although it could be a reduction in either efficacy or potency. These potency SARs at $R_1$ are in good correspondence with the predicted binding affinities from QSAR modelling[35,36].

At the 3 position ($R_3$), the parents with a hydrogen were more potent than the metabolites with a hydroxy, but the efficacy relationship was unclear. Based on the prodrug cinazepam (**7**) with a bulky group (butanedioic acid) at $R_3$, a bulky group results in a significant reduction in potency but not efficacy; however, this is only one comparison, so more BZDs with a bulky group at $R_3$ should be tested to confirm this SAR. All of these potency relationships at $R_3$ were the same as the predicted binding affinities from QSAR modelling, except the metabolites 3-hydroxyphenazepam (**4**) and nifoxipam (**22**) showed greater activity (larger log(1/c) values) than their parents phenazepam (**26**) and desmethylflunitrazepam (**12**), respectively, from the QSAR[35,36]. Overall, these results suggest that QSAR modelling can be a valuable tool to estimate the activity of DBZDs as new drugs emerge; however, the identified inconsistencies demonstrate the importance of experimental testing for evaluating the actual activity. The experimental data in this study could be used in the future to help improve the current QSAR models.

All the metabolites in this study showed GABA$_A$ activity, sometimes even greater than the parent drug, which demonstrates that metabolites of (D)BZDs are important for understanding the overall pharmacological activity of these drugs. This is especially true given BZDs have metabolic half-lives between 5 and more than 24 h and therefore can have long-acting metabolites that accumulate during repeat dosing[37,38]. Therefore, in the future, the in vitro activity of more (D)BZD metabolites should be explored to further elucidate the SARs of metabolism and the overall pharmacological activity of these drugs.

In addition, although the BZD prodrugs metabolize into a more potent and typically more efficacious primary metabolite and other potentially active metabolites, they can still have long metabolic half-lives. For example, gidazepam (**18**) has a half-life of over 24 h[29] and cinazepam (**7**) has a half-life of 15–17 h[39]. This means the activity of the prodrugs themselves is still important for considering the effects and risks of harm from the use of these drugs. On the other hand, open-ring BZD prodrugs like rilmazafone (**53**) metabolize more quickly, where little to no rilmazafone (**53**) was detected in plasma within 20–60 min of administration in three human subjects[40] and no rilmazafone (**53**) was detected in blood or urine samples from forensic cases[26]. Therefore, the activity of open-ring BZD prodrugs will have limited

impact on the overall pharmacological activity, so the major metabolites should be the focus of activity examinations of open-ring prodrugs in the future. For that reason, only the closed-ring analogs of three new open-ring prodrugs reported to the EU EWS in September 2024 were examined in this study: nordazepam (**24**; metabolite of noravizafone desglycyl)[41], clonazepam (**8**; metabolite of clonazafone desglycyl)[42], and delorazepam (**10**; metabolite of diclazafone desglycyl)[43].

In addition to the overall activity at the GABA$_A$ receptor, it was found that some (D)BZDs may bind to other binding interfaces on the GABA$_A$ receptor. As can be seen in the concentration-response curves in Fig. 5 and Supplementary Information Sections 2–6, 13 (D)BZDs exhibited a typical sigmoidal concentration-response curve (e.g., norfludiazepam (**25**) and clonazepam (**8**) in Fig. 4), but 29 had bell-shaped concentration-response curves with inhibition of the activity at higher concentrations (e.g., diazepam (**13**), delorazepam (**10**) in Fig. 5) and 3 had more complex relationships with biphasic potentiation/inhibition/potentiation (nitrazolam (**37**), clonazolam (**33**), and adinazolam (**29**) in Supplementary Fig. 4). Bell-shaped curves have been found in previous studies on the GABA$_A$ receptor expressed in oocytes with other drugs (e.g., methaqualone, propofol) using two-electrode voltage clamp[44–46]. The reason for the bell-shaped and more complex concentration-response curves is unclear, but it was suspected that it could be due to the drug interacting with other binding sites on the GABA$_A$ receptor, leading to inhibition. If this were the case, then it was expected that the inhibition would still be present when the BZD antagonist, flumazenil (**54**), was applied; however, this was not found. Instead, most (D)BZDs had a complete reduction of activity in the presence of flumazenil (**54**), suggesting their observed activity is largely mediated by the BZD interface. Six DBZDs (2'-bromonordazepam (**2**), clobromazolam (**32**), clonazolam (**33**), deschloroclotizolam (**42**), difludiazepam (**15**), and thionordazepam (**49**)) had some remaining potentiation in the presence of flumazenil (**54**) (Fig. 6B–G), which could be from the binding of these DBZDs at other interfaces on the GABA$_A$ receptor or from competing with flumazenil (**54**). The remaining activation of deschloroclotizolam (**42**) in the presence of flumazenil (**54**) even had a bell-shaped curve (Fig. 6E), further indicating that the shape of the concentration-response curves is likely not a result of inhibition mediated by another binding interface.

Overall, halogenation at the 4 position was found to be the most significant SAR in this study, as it reversed the activity, producing NAMs. 4'-fluorodiazepam (**6**) and 4'-chlorodiazepam (**5**) showed inhibition with and without the presence of flumazenil (**54**), indicating that the observed NAM from these DBZDs is not mediated by the BZD interface. This is consistent with previous studies of 4'-chlorodiazepam (**5**) in cortical neurons of neonatal rats, mouse spinal cord, and dorsal root ganglion neurons that also found flumazenil (**54**) did not antagonize its action[47,48]. 4'-chlorodiazepam (**5**) has been identified as a ligand of TSPO, formerly known as the peripheral benzodiazepine receptor[49], which is implicated in fundamental cell processes like steroid biosynthesis, cell proliferation and differentiation, immunomodulation, and apoptosis[50]. 4'-chlorodiazepam (**5**) has been found to be neuroprotective against amyloid-beta[51], analgesic for neuropathic pain[52,53], antidepressant with steroid synthesis suggested as an underlying mechanism[54], anti-inflammatory through suppression of mast cells[34] and regulation of T-cells[55], cardioprotective with multiple mechanisms identified[56–59], and induce apoptosis in cancer cells[60–63]. These effects have largely been suggested to be as a result of its interaction with TSPO rather than GABA$_A$, but the effects of its observed inhibition of GABA$_A$ are unclear, and the GABA$_A$ binding interface has yet to be identified.

2,4'-dichlorodiazepam (**3**) and 4'-chloro deschloroalprazolam (**28**) were also found to produce inhibition that was not antagonized by flumazenil (**54**), but also potentiation that was antagonized by flumazenil (**54**), resulting initially in the appearance of no significant activity. This indicates that the inhibition is likely mediated by the same GABA$_A$ binding interface as 4'-fluorodiazepam (**6**) and 4'-chlorodiazepam (**5**), while the potentiation is mediated by the BZD interface. This combination of potentiation and inhibition could produce a combination of the effects of 4'-chlorodiazepam (**5**) and standard BZDs (i.e., diazepam (**13**), alprazolam (**30**)) or no overall

effects from the GABA$_A$ receptor. Previous studies and patents indicate that similar to 4'-chlorodiazepam (**5**), 2,4'-dichlorodiazepam (**3**) is a ligand of TSPO[64,65] and can be used to treat tumors and cancer[66,67], inflammation[68], CNS trauma or disease[69], cellular damage from exposure to tobacco smoke or oxidative stress[70] and adverse effects of methamphetamine use and HIV infection, including medical and behavioral consequences of methamphetamine use, HIV-associated neurodegenerative disorder, and neuroinflammatory response[71]. Although these effects have largely been suggested to be a result of its interaction with TSPO rather than GABA$_A$, this data suggests that, despite both potentiating and inhibiting GABA$_A$ from different binding interfaces, 2,4'-dichlorodiazepam (**3**) still produces positive effects.

As it is the most significant SAR, para-halogenation of the phenyl ring should be explored further by examining the GABA$_A$ activity of more para halogenated 1,4-BZDs and 1,3,4-triazolos and BZDs with other base structures like the thienotriazolos. Future research should also aim to identify the GABA$_A$ binding interface mediating the inhibition observed for these para halogenated BZDs. In addition, since there have been many positive effects found for 4'-chlorodiazepam (**5**) and 2,4'-dichlorodiazepam (**3**) correlated to interaction with TSPO, it possible that 4'-fluorodiazepam (**6**) and other para halogenated 1,4-BZDs and 4'-chloro deschloroalprazolam (**28**) and other para halogenated 1,3,4-triazolos may produce similar effects. Therefore, these BZDs should be examined for binding with TSPO, and the effects of these BZDs resulting from the GABA$_A$ inhibition versus TSPO binding should be explored.

## Conclusions

In this study, there was large variation observed in the pharmacological activity at GABA$_A$ for prescription and DBZDs, demonstrating the importance of testing new drugs as they emerge on the recreational market to inform harm reduction measures and drugs legislation. As summarized in Fig. 7, there were several significant SARs identified that influence efficacy and/or potency. This offers insights into how chemical modifications may alter pharmacodynamics and could help with predicting the activity of DBZDs that emerge in the future and the development of new drugs for therapeutic use. In the future, the pharmacological activity of more structurally diverse BZDs, particularly more thienotriazolos, should be examined to further elucidate the SARs identified in this study.

Examination of the binding interactions of the DBZDs using the BZD antagonist flumazenil (**54**) revealed that the majority of the activity of most DBZDs is mediated by the BZD $\alpha^+/\gamma2^-$ interface. However, there were six DBZDs that had remaining potentiation in the presence of flumazenil (**54**), indicating they are either competitive with flumazenil (**54**) or have some potentiation mediated through another binding interface. In addition, the inhibitory activity of the DBZDs with halogenation at position 4 on the phenyl ring ($R_7$) was found to be mediated independently of the BZD $\alpha^+/\gamma2^-$ interface. This demonstrates that even small changes to the BZD structure can alter the interactions with the GABA$_A$ binding interfaces, which may impact the pharmacological and physical effects of the drugs. Further examination of these DBZDs to confirm their interaction with another GABA$_A$ interface and identify the interface(s) involved could help advance the current understanding of the mechanisms of action of BZDs and the binding interfaces of GABA$_A$.

## Methods
### Chemicals and reagents
A Chinese hamster ovary (CHO) $\alpha_1\beta_2\gamma_2$ GABA$_A$ cell line that stably expresses the human GABA$_A$ receptor (B'SYS GmbH, Witterswil, Switzerland; RRID:CVCL_C0XQ) was used. HAM's F-12 nutrient mix, Dulbecco's phosphate buffered saline (DPBS) without $Ca^{2+}$ and $Mg^{2+}$, penicillin-streptomycin (10,000 U/mL), and sucrose (99%) were obtained from Thermo Fisher Scientific (Waltham, Massachusetts, USA). Ex-cell animal-component free (ACF) CHO medium was obtained from Merck (Darmstadt, Germany). Detachin™ cell detachment solution was purchased from AMSBIO (Abingdon, UK). Heat-inactivated fetal bovine serum (FBS),

HEPES, hygromycin B, puromycin, soybean trypsin inhibitor, egtazic acid (EGTA), calcium chloride (CaCl$_2$), magnesium chloride (MgCl$_2$), potassium hydroxide (KOH), potassium chloride (KCl), sodium hydroxide (NaOH), adenosine 5′-triphosphate disodium salt hydrate (Na$_2$ATP), dimethyl sulfoxide (DMSO), GABA, and D-(+)-glucose were obtained from Sigma Aldrich (Stockholm, Sweden). Zeocin was obtained from InvivoGen (San Diego, California, USA). Sodium chloride (NaCl) was obtained from VWR International (Radnor, Pennsylvania, USA).

### Reference standards
Reference standards for 3-hydroxyphenazepam (**4**), 4′-chloro deschloroalprazolam (**28**), 4′-fluorodiazepam (**6**), bromazolam (**31**), brotizolam (**40**), cinazepam (**7**), clobromazolam (**32**), clotizolam (**41**), delorazepam (**10**), desalkylgidazepam (**11**), deschloroclotizolam (**42**), difludiazepam (**15**), flualprazolam (**34**), flubrotizolam (**45**), fluclotizolam (**46**), fluetizolam (**47**), gidazepam (**18**), methylclonazepam (**21**), nitrazepam (**23**), thionordiazepam (**49**), and tofisopam (**50**) were obtained from Cayman Chemical (Ann Arbor, MI, USA). Reference standards for 2′-bromodiazepam (**1**), 2′-bromonordazepam (**2**), 2,4′-dichlorodiazepam (**3**), adinazolam (**29**), bentazepam (**52**), clonazolam (**33**), cloniprazepam (**9**), deschloroetizolam (**43**), flubromazepam (**16**), flubromazolam (**35**), flunitrazolam (**36**), lorazepam (**19**), meclonazepam (**20**), metizolam (**48**), nifoxipam (**22**), nitrazolam (**37**), phenazepam (**26**), and pyrazolam (**38**) were purchased from Chiron AS (Trondheim, Norway). The reference standards for desmethylflunitrazepam (**12**), diclazepam (**14**), fludiazepam (**17**), and nordazepam (**24**) were purchased from LGC Ltd (Teddington, UK). Reference standards for 4′-chlorodiazepam (**5**), alprazolam (**30**), bretazenil (**51**), clonazepam (**8**), diazepam (**13**), etizolam (**44**), norfludiazepam (**25**), and temazepam (**27**) were purchased from Sigma Aldrich. Flumazenil (**54**) was obtained from VWR International (Stockholm, Sweden). The rilmazafone (**53**) reference standard was purchased from Chemtronica (Sollentuna, Sweden), and its main metabolite, rilmazolam (**39**), was synthesized inhouse as described previously[26].

### Buffers
The Sophion extracellular (EC) solution 0.0.0 stock solution was prepared in ultra-high purity water obtained using a Milli-Q water purification system with 2 mM CaCl$_2$, 1 mM MgCl$_2$, 10 mM HEPES, 145 mM NaCl, and 4 mM KCl and was filtered. On the day of the experiments, the EC solution was prepared from the stock by adding 1 M glucose (final concentration 10 mM) and then adjusting the pH to 7.40 ± 0.01 using NaOH and the osmolarity to 305–308 mOsm with sucrose.

The Sophion intracellular (IC) solution 0.0.0 stock solution was prepared in ultra-high purity water with 5.374 mM CaCl$_2$, 1.75 mM MgCl$_2$, 10 mM HEPES, 10 mM EGTA, and 120 mM KCl and was filtered. On the day of the experiments, the IC solution was prepared from the stock by adding Na$_2$ATP (final concentration 4 mM) and then adjusting the pH to 7.20 ± 0.01 using KOH and the osmolarity to 305–308 or 310–313 mOsm with sucrose. The osmolarity of the IC solution was later increased to 310–313 mOsm to improve the stability of the cells in the assay. This did not have any impact on the calculated efficacy or potency (see Supplementary Fig. 3), so all data are comparable regardless of the IC solution used.

### In vitro biological activity at GABA$_A$ receptor
CHO cells stably expressing the α$_1$β$_2$γ$_2$ GABA$_A$ receptor were maintained in a humidified atmosphere at 37 °C and 5% CO$_2$ in Ham's F-12 nutrient mix supplemented with 10% heat-inactivated FBS, 1% penicillin and streptomycin, 250 μg/mL hygromycin B, 5 μg/mL puromycin, and 100 μg/mL zeocin. Cells were incubated at 30 °C the night before an experiment to improve expression of the receptor and increase the electrical current. To perform the assays, cells were detached using Detachin™ (5 min, 37 °C), resuspended in ACF CHO medium supplemented with 25 mM HEPES, 100 U/mL penicillin/streptomycin, and 0.04 mg/mL soybean trypsin inhibitor, then counted. Cells were added to a lidless microtube at the QPatch so the QPlate cell density would be about $10.5 \times 10^6$ per well.

The QPatch II (Sophion Bioscience A/S, Ballerup, Denmark), an automated patch-clamp system, was used to obtain data recordings of the GABA$_A$ assay. QPlate 48X (Sophion Bioscience A/S), 48-well plates for multi-hole recordings with integrated flow channels for intracellular (IC) and extracellular (EC) solutions and integrated electrodes, were used. The QPlate 48X has 10 holes per measurement site, allowing for the testing of 10 cells within each well, with the final recordings an average of all cells. For recordings, the QPatch first adds IC solution to the IC flow channels, followed by EC solutions to the EC flow channels. The cells, which were centrifuged and then resuspended in EC solution, were injected, and increasing suction applied to pull the cells into the holes where the cell wall was gently ruptured for whole cell recordings, as depicted in Fig. 1B. Electrodes in the EC and IC flow channels recorded the electrical signal as it changed due to chloride influx from activation of the receptor. An example of the electrical currents recorded with this method from one cell for increasing concentrations of diazepam (**13**) is shown in Fig. 1C, along with the resulting concentration-response curve for diazepam (**13**) from a minimum of four independent experiments ($n \geq 4$).

The assay protocols were prepared based on the details provided in "basic protocol 1" in Schupp et al.[72]. The whole-cell protocol used had a holding potential of −90 mV and −20 mbar. The chip validation required a resistance between 0.5 and 10 MΩ. The cell positioning time was 30 s with a positioning pressure of -100 mbar. For whole-cell, five suction pulses were used starting at a first pulse amplitude of −200 mbar with a −50 mbar increment between pulses, pulse duration of 1000 ms, and a pulse period of 10,000 ms.

The cells were clamped to −90 mV throughout the recordings. The minimum seal resistance was 1 MΩ, liquid dispense delay of 3000 ms, data acquisition time of 14,000 ms, cut-off frequency of 3000 Hz, sampling frequency of 10,000 Hz, and filter order of 8. The signal was filtered at 3000 Hz (filter order 8) using a Bessel filter.

**Application protocol**. A total of 12 recordings (5 s) were performed for each cell (six test solution concentrations with three vehicle (2 μM GABA) recordings before and after the concentration series for controls). In total, 10 μL was added to the EC flow channels (5 μL applied twice, with a 3 s interval between applications). Five seconds after each application, a pipette washing step was performed. The liquid exposure duration ranged from 40 to 300 s, and the total experiment duration did not exceed 4000 s.

**Experiments for BZD concentration-response curves**. During optimization of the assay, it was found that the BZDs take time to bind and lead to the increase in chloride influx; therefore, it was necessary to pre-incubate the cells with each concentration of the BZD before adding GABA in order to see the full effect of each BZD concentration on the chloride influx (see Supplementary Fig. 2). The baseline concentration of GABA chosen for the experiments was 2 μM GABA, about 10% of the $I_{max}$ (see Supplementary Fig. 1).

Compound dilutions were prepared in DMSO in 96-well microtiter plates (MTP-96) with glass inserts (Sophion Bioscience A/S). For a standard plate layout, the first row was a set of vehicle control (0.3% DMSO), and the following seven rows each had one test compound with two sets of six increasing concentrations. The first six columns (first set of the six concentrations) were used for pre-incubation and did not contain GABA, while the last six columns (second set of the six concentrations) were used for the main application and contained 2 μM GABA. For determining a BZD concentration-response curve, concentrations ranged between 0.96 and 3000 nM with a 1:5 dilution. For high potency BZDs, additional concentrations as low as $1.54 \times 10^{-3}$ nM were tested. Compound dilutions were individually prepared (not as a serial dilution) so that all wells contained 0.3% DMSO. The data used to generate the concentration-response curves for all 53 BZDs are provided in Supplementary Data 1.

### Experiments for examining the binding interface of DBZDs

To try to determine if the $\alpha^+/\gamma_2^-$ interfaces (BZD binding site) is the main binding site for DBZDs, the BZD antagonist, flumazenil (**54**), was used to try to block the BZD binding site. For these experiments, flumazenil (**54**) was added to both the pre-incubation and main application sets along with the (D)BZD. First, in order to determine the concentration of flumazenil (**54**) to use for the screening experiments, a concentration-response curve with flumazenil (**54**) and diazepam (**13**) was determined using increasing concentrations of flumazenil (**54**) (0.96–3000 nM) with 98 nM of diazepam (**13**; approximately $EC_{80}$). From this, it was determined that concentrations of flumazenil (**54**) above 120 nM blocked the activity of diazepam (**13**). Next, two concentration-response curves were determined with increasing concentrations of diazepam (**13**; 0.96–3000 nM) with 120 or 600 nM flumazenil (**54**). Both 120 and 600 nM flumazenil (**54**) were found to block the activity of all concentrations of diazepam (**13**), but to try to better ensure successful blocking of the BZD interface with DBZDs, 600 and 3000 nM of flumazenil (**54**) were chosen to be used for the screening experiments.

For the binding interface experiments, the compound plate layout was similar to the one used for BZD concentration-response curves, but the vehicle control was 600 or 3000 nM flumazenil (**54**) and 0.3% DMSO. For the screening, only two concentrations of each DBZD were used, one around the $EC_{50}$ and one around the $EC_{80}$. Therefore, per row of the MTP plate, there were three test compounds with two concentrations each, still with the pre-incubation and main application sets. As it had been observed that some DBZDs are difficult to wash away from the receptor, at least the first two compounds tested needed to result in no activity in order to successfully confirm the BZD interface for all DBZDs tested in a row. Therefore, to improve the chances of success, compounds were organized in each row so the first compound had a low potency (high $EC_{50}$), the second had a medium potency, and the third had a high potency (low $EC_{50}$). In the event that the first or second compound in the row resulted in any activity, all following compounds were re-run. If the activity persisted, a full concentration-response curve was determined with 3000 nM flumazenil (**54**) and increasing concentrations of the compound of interest (0.96–3000 nM). Full concentration-response curves were also determined with 3000 nM flumazenil (**54**) and increasing concentrations of the compound of interest (0.96–3000 nM) for all DBZDs exhibiting no activity or inhibitory activity. The data from the screening with flumazenil and the data used to generate concentration-response curves in the presence of flumazenil for select DBZDs are provided in Supplementary Data 2.

### Data analysis

The recordings were analyzed in the Sophion Analyzer 7.0 software (Sophion Bioscience A/S), followed by GraphPad Prism (Version 10.0.2). Data were normalized to an average of the first three vehicle controls from each well and then by an average of the complete set of vehicle control runs on the same QPlate. The data were normalized so the baseline GABA (2 μM) activity was at an efficacy of 1. The concentration-response curves were generated, and $EC_{50}$ and $E_{max}$ values for potentiating BZDs were calculated by curve fitting via nonlinear regression (three-parameter logistic fit) with the following constraints: bottom is a constant equal to 1, and $EC_{50}$ must be greater than 0. The concentration-response curves were generated, and $IC_{50}$ and $I_{max}$ values for inhibiting BZDs were calculated by curve fitting via nonlinear regression (three-parameter logistic fit) using the following constraints: bottom must be between zero and one, top is constant equal to one, and the $IC_{50}$ must be greater than 0. For compounds that have a reduction in current at higher concentrations, leading to a bell curve, the concentrations above the first point of current reduction were excluded from the curve fitting. Results are represented as the normalized electric current derived from a minimum of four independent experiments ($n \geq 4$). Brown-Forsythe and Welch ANOVA tests ($\alpha = 0.05$) were performed in GraphPad Prism to compare the $EC_{50}$ and $E_{max}$ values between compounds.

### Reporting summary

Further information on research design is available in the Nature Portfolio Reporting Summary linked to this article.

### Data availability

The authors declare that all data supporting the findings of this study are available within the paper and its supplementary information files.

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

## Acknowledgements

The authors thank Sophion Bioscience A/S (Ballerup, Denmark), particularly Naja Møller Sørensen, for their assistance and support in the development of the assay used in this study. They also thank B'SYS GmbH (Witterswil, Switzerland) for supplying the GABA$_A$ CHO-K1 cell line used. This study received funding from the European Union's Horizon 2020 research and innovation program, Sweden's Innovation Agency (grant number 2019-03566), the Strategic Research Area in Forensic Sciences (Strategiområdet forensiska vetenskaper) at Linköping University, Kungliga Vetenskapsaka-demien (The Royal Swedish Academy of Sciences; Medical Sciences 2024, ME2024-0012), Svenska Sällskapet för Medicinsk Forskning (Swedish Society for Medical Research, SSMF; Postdoctoral grant 2024, PG-24-0356-H-01), and the Chemical Biology Consortium Sweden (CBCS). We acknowledge the support of the Chemical Biology Consortium Sweden (CBCS), the node in Linköping University, for funding Amaia Jauregi Miguel and Nina Ottosson. CBCS is a national research infrastructure funded by the Swedish Research Council (dr.nr.2021-00179) and SciLifeLab.

## Author contributions
Conceptualization: C.N. and H.G.; Methodology: C.N., A.J.M. and N.O.; Investigation: C.N., A.J.M. and N.O.; Formal analysis: C.N. and N.O.; Visualization: C.N.; Supervision: S.L. and H.G.; Writing—original draft: C.N.; Writing—review and editing: all.

## Funding

## Competing interests
The authors declare no competing interests.
