## [Transparent Peer Review file · Communications Chemistry]

In vitro γ -aminobutyric acid A (GABAA) receptor activity and binding interactions at the α_+ / γ_2 - interface of 53 prescription and designer benzodiazepines

Corresponding Author: Dr Caitlyn Norman

Version 0:

Reviewer comments:

Reviewer #1

(Remarks to the Author)

The authors describe here their extensive study of the activity of 53 benzodiazepines (BZDs), 42 of which were designer BZDs (DBZDs), at $\alpha_1\beta_2\gamma_2$ receptors in vitro. All tested compounds were positive allosteric modulators acting at the α_+ / γ_2 -interface (except for compounds with substituents at the 4'-position of the BZD phenyl ring). The results show that, like the prescribed BZDs, DBZDs also act by increasing GABAA receptor activity. The large number of compounds tested provides detailed information on how modification of the BZD structure affects their pharmacodynamics.

The experiments are described in detail from the setting up of the method, allowing for their reproducibility.

ANOVA and t-tests were used as appropriate. Error bars are NOT specified in Supplementary Figures S1.1.B., S1.2.C. and S1.3. Otherwise, error bars are well defined. Statistical significance is set at a P value of 0.05. Confidence intervals are calculated at a 95% confidence level.

The interpretation of the data and the conclusions drawn are robust, valid and reliable.

The manuscript is generally written in excellent scientific English. However, the HALOGEN GROUPS of BZD compounds are incorrectly expressed as halides (chloride, bromide, fluoride). These atoms are not in ionized form. Authors should use either chlorine, bromine, etc. or chlorine, bromine group, chlorine substituent, etc.

Reviewer #2

(Remarks to the Author)

The MS is enough interesting. I suggest to add a small limitation. GABAA is not alone target of BZD. I think the authors should mention that part of effects of BZD may related to interaction with translocator protein but study of DBZD with this target was out of scope of this paper.

Version 1:

Reviewer comments:

Reviewer #1

(Remarks to the Author)

The points raised in the previous round of review have been satisfactorily addressed in the revised manuscript and in the supplementary material.

Reviewer #2

(Remarks to the Author)
Congrats!

To the Editors and Reviewers at Communications Chemistry,

We thank the Editor and Reviewers for the detailed reading and for the overall positive assessment of our manuscript, COMMSCHEM-25-1213-T. Below we have outlined a detailed response to all the comments that were made. We feel that we have been able to adequately address all comments and are grateful to the Editor and Reviewers for helping us to further improve our manuscript. We think the manuscript has been improved by the revision and we now hope it is suitable for publication in Communications Chemistry.

Editorial comments:

Please ensure that the following requirements are met, and that any other relevant checklists are completed and uploaded under the 'Related Manuscript file' type with the revised article.

Chemical and biomolecular materials: Characterization of chemical and biomolecular materials

In order to comply with the characterization guidelines, bold Arabic numerals have been added for all compounds throughout the text and tables in the manuscript.

NMR spectra: Requirements for the provision of NMR spectra

This is not applicable as we have no NMR spectra in this manuscript.

Life sciences reporting summary: Reporting requirements for life sciences research

This form has been filled out and provided in the submission.

In the event that your manuscript is accepted, we will provide detailed guidance on our journal policies and formatting. You may however wish to ensure that the manuscript broadly complies with our house style at this stage. See our style and formatting guide (<https://www.nature.com/documents/commsj-phys-style-formatting-guide-accept.pdf>) and checklist (<https://www.nature.com/documents/commsj-phys-style-formatting-checklist-article.pdf>) for reference.

The manuscript and supplementary information have been checked for compliance with the journal policies and formatting. This includes the following changes:

- References to display items in the Supplementary Information have been added to the main text, as required.
- A brief title has been added to figure captions that do not contain reference to specific panels.
- Abbreviations are defined in the display item captions.

Data availability statements and data citations policy: All Communications Chemistry manuscripts must include a section titled "Data Availability" at the end of the Methods section or main text (if no Methods). More information on this policy, and a list of examples, is available at <http://www.nature.com/authors/policies/data/data-availability-statements-data-citations.pdf>.

We have added the following data availability statement: "The authors declare that all data supporting the findings of this study are available within the paper and its supplementary information files."

Reviewers' comments:

Reviewer #1 (Remarks to the Author):

The authors describe here their extensive study of the activity of 53 benzodiazepines (BZDs), 42 of which were designer BZDs (DBZDs), at $\alpha 1\beta 2\gamma 2$ receptors in vitro. All tested compounds were positive allosteric modulators acting at the $\alpha +/\gamma 2$ - interface (except for compounds with substituents at the 4'-position of the BZD phenyl ring). The results show that, like the prescribed BZDs, DBZDs also act by increasing GABAA receptor activity. The large number of compounds tested provides detailed information on how modification of the BZD structure affects their pharmacodynamics.

We thank the reviewer for their overall positive assessment of our manuscript.

The experiments are described in detail from the setting up of the method, allowing for their reproducibility.

We thank the reviewer for their positive assessment of our method details.

ANOVA and t-tests were used as appropriate. Error bars are NOT specified in Supplementary Figures S1.1.B., S1.2.C. and S1.3. Otherwise, error bars are well defined. Statistical significance is set at a P value of 0.05. Confidence intervals are calculated at a 95% confidence level.

Thank you for your thorough evaluation of our statistical analyses. As requested, we have specified in the headings for Supplementary Figures S1.1B, S1.2C, and S1.3 that "Errors bars indicate standard error to the mean (SEM)".

The interpretation of the data and the conclusions drawn are robust, valid and reliable.

We thank the reviewer for their positive assessment of our data interpretation and conclusions.

The manuscript is generally written in excellent scientific English. However, the HALOGEN GROUPS of BZD compounds are incorrectly expressed as halides (chloride, bromide, fluoride). These atoms are not in ionized form. Authors should use either chlorine, bromine, etc. or chlorine, bromine group, chlorine substituent, etc.

Thank you for the suggestion. We have changed all uses of the halides (chloride, fluoride, bromide, hydride) to the non-ionized atoms (chlorine, fluorine, bromine, hydrogen) when discussing the benzodiazepine compounds (there are some uses of chloride when discussing the GABA receptor chloride flux, which we did not change as those are correct).

Reviewer #2 (Remarks to the Author):

The MS is enough interesting. I suggest to add a small limitation. GABAA is not alone target of BZD. I think the authors should mention that part of effects of BZD may related to interaction with translocator protein but study of DBZD with this target was out of scope of this paper.

We thank the reviewer for their overall positive assessment of our manuscript. We agree that ideally other targets of the BZDs would be examined to get a better picture of their effects and toxicity, particularly other GABA_A isoforms and TSPO. In the discussion, we have added the following sentence addressing other possible targets of benzodiazepines:

Page 11, Lines 27-29: “It may also be due to BZDs interacting with other targets, including other GABA_A isoforms and mitochondrial translocator protein 18kDa (TSPO), which were outside of the scope of this study to examine.”